# Sulfonylpiperazine compounds prevent *Plasmodium falciparum* invasion of red blood cells through interference with actin-1/profilin dynamics

**Madeline G. Dans**[1,2,3]*, **Henni Piirainen**[4], **William Nguyen**[3], **Sachin Khurana**[3], **Somya Mehra**[1], **Zahra Razook**[1,2], **Niall D. Geoghegan**[3], **Aurelie T. Dawson**[3], **Sujaan Das**[5], **Molly Parkyn Schneider**[1], **Thorey K. Jonsdottir**[1,6], **Mikha Gabriela**[1,2], **Maria R. Gancheva**[7], **Christopher J. Tonkin**[3], **Vanessa Mollard**[8], **Christopher Dean Goodman**[8], **Geoffrey I. McFadden**[8], **Danny W. Wilson**[7], **Kelly L. Rogers**[3], **Alyssa E. Barry**[1,2], **Brendan S. Crabb**[1,6], **Tania F. de Koning-Ward**[2], **Brad E. Sleebs**[3], **Inari Kursula**[4,9], **Paul R. Gilson**[1,6]*

1 Burnet Institute, Melbourne, Victoria, Australia, 2 School of Medicine and Institute for Mental and Physical Health and Clinical Translation, Deakin University, Waurn Ponds, Victoria, Australia, 3 Walter and Eliza Hall Institute, Parkville, Victoria, Australia, 4 Faculty of Biochemistry and Molecular Medicine, University of Oulu, Oulu, Finland, 5 Ludwig Maximilian University, Faculty of Veterinary Medicine, Munich, Germany, 6 Department of Microbiology and Immunology, The University of Melbourne, Parkville, Victoria, Australia, 7 Research Centre for Infectious Diseases, The University of Adelaide, Adelaide, Australia, 8 School of Biosciences, The University of Melbourne, Parkville, Victoria, Australia, 9 Department of Biomedicine, University of Bergen, Bergen, Norway

* dans.m@wehi.edu.au (MGD); paul.gilson@burnet.edu.au (PRG)

## Abstract

With emerging resistance to frontline treatments, it is vital that new antimalarial drugs are identified to target *Plasmodium falciparum*. We have recently described a compound, MMV020291, as a specific inhibitor of red blood cell (RBC) invasion, and have generated analogues with improved potency. Here, we generated resistance to MMV020291 and performed whole genome sequencing of 3 MMV020291-resistant populations. This revealed 3 nonsynonymous single nucleotide polymorphisms in 2 genes; 2 in *profilin* (N154Y, K124N) and a third one in *actin-1* (M356L). Using CRISPR-Cas9, we engineered these mutations into wild-type parasites, which rendered them resistant to MMV020291. We demonstrate that MMV020291 reduces actin polymerisation that is required by the merozoite stage parasites to invade RBCs. Additionally, the series inhibits the actin-1-dependent process of apicoplast segregation, leading to a delayed death phenotype. In vitro cosedimentation experiments using recombinant *P. falciparum* proteins indicate that potent MMV020291 analogues disrupt the formation of filamentous actin in the presence of profilin. Altogether, this study identifies the first compound series interfering with the actin-1/profilin interaction in *P. falciparum* and paves the way for future antimalarial development against the highly dynamic process of actin polymerisation.

**Data Availability Statement:** Genomic sequencing data is available from European Nucleotide Archive;

accession number PRJEB55647. Source data are available in S1 Data.

**Funding:** This work was supported by the National Health and Medical Research Council (2001073 to P.R.G and W.N) and (119780521 to B.S.C), the Victoria Operational Infrastructure Support Programs received by the Burnet Institute and Walter and Eliza Hall Institute, the Academy of Finland (322917 to I.K and H.P), the Sigrid Jusélius Foundation (to I.K.) and the Hospital Research Foundation (to D.W.W). This work was also funded by an Australian Government Research Training Program Scholarship (to M.G.D), a University of Melbourne Research Scholarship (to T.K.J), an Ellen Corin Fellow (to B.E.S) and an National Health and Medical Research Council Senior Research Fellowship (1136300 to TdK-W). The funders had no role in study design, data collection and analysis, decision to publish, or preparation of the manuscript.

**Competing interests:** The authors have declared that no competing interests exist.

**Abbreviations:** ACP-GFP, green fluorescent protein-tagged acyl carrier protein; ADF1, actin-depolymerisation factor 1; BSA, bovine serum albumin; CytD, cytochalasin D; F-actin, filamentous actin; FBS, foetal bovine serum; G-actin, globular actin; gRNA, guide RNA; hDHFR, human dihydrofolate resistance; HsPFNI, *Homo sapiens* profilin I; LDH, lactate dehydrogenase; MoA, mechanism of action; PBS, phosphate buffered saline; PfACT1, *P. falciparum* actin-1; PfPFN, *P. falciparum* profilin; PV, parasitophorous vacuole; RBC, red blood cell; RLU, relative light unit; SAR, structure activity relationship; SNP, single nucleotide polymorphism; TgPFN, *Toxoplasma gondii* profilin; WT, wild-type.

# Introduction

Malaria is a devastating parasitic disease that caused approximately 619,000 deaths in 2021, an upward trend from 2020's figure of 558,000 due to COVID-19-related service disruption [1]. The majority of these deaths were a result of infection with *Plasmodium falciparum*, which causes widespread disease across sub-Saharan Africa. The alarming increase in deaths, combined with the spread of mutations conferring resistance to frontline antimalarials [2–4], highlights the urgent need to find new compounds with unique mechanisms of action (MoA) to combat this deadly parasite.

The red blood cell (RBC) stage of *Plasmodium* infection within the human host leads to the exponential growth of the parasite and the symptoms of the disease. Within RBCs, parasites develop within a parasitophorous vacuole (PV) in a series of stages from rings to trophozoites and, finally, schizonts. During schizogony, daughter merozoites are formed, which eventually egress from the RBC to reinfect new RBCs. The invasion of RBCs by merozoites is a complex and finely tuned process that involves a multitude of unique signalling cascades and protein–protein interactions [5]. These events must occur within minutes of a merozoite egress to allow successful RBC internalisation, and, therefore, this process represents a prime opportunity against which to develop therapeutics [6–9]. Administering an invasion inhibitor in combination with a drug targeting intracellular parasite processes would span the entire asexual blood stage of infection and could effectively prevent parasite proliferation [8].

One unique process required for parasite invasion of RBCs is the engagement of an actomyosin motor complex, termed the glideosome, a mechanism that is shared between apicomplexan parasites. In the glideosome, a single-headed class XIV myosin A, MyoA, is tethered to the inner membrane complex of the merozoite via its 2 light chains and several glideosome-associated proteins [10–12]. MyoA produces the force required for gliding motility by walking along filamentous actin [13–16]. Actin filaments, in turn, are linked to surface exposed adhesin proteins via the glideosome-associated connector [17]. A ring of adhesions, termed the tight junction, is then formed between the apical tip within the merozoite and the RBC, and the force produced by the MyoA power stroke propels the merozoite into the RBC through the established tight junction [18,19]. This movement translocates the tight junction from the apical to the posterior end of the parasite and results in the RBC membrane enveloping the merozoite that later forms the PV [5,20,21].

The generation of filamentous actin (F-actin) is required for both gliding motility and RBC invasion by the parasite [22,23]. F-actin is formed through the incorporation of subunits of globular actin (G-actin)-ATP at the barbed end. To stabilise the growing filament, hydrolysis of G-actin-ATP occurs to generate G-actin-ADP and inorganic phosphate ($P_i$) [24–26]. For filament disassembly and G-actin turnover, the release of $P_i$ at the pointed end destabilises the F-actin, resulting in the disassociation of G-actin-ADP [27,28]. The continuous growth at the barbed end and shortening at the pointed end of the filament is referred to as actin treadmilling [25] and is a process that is tightly regulated by a plethora of actin-binding regulatory proteins, many of which are absent in apicomplexan parasites [14,29]. However, one key protein present across eukaryotes from Apicomplexa to Opisthokonts is profilin, a sequester of G-actin [30]. The role of profilin is to maintain a pool of polymerizable actin monomers and to catalyse the exchange of ADP to ATP in the monomers [31,32]. Profilin then delivers these G-actin-ATP to formin, a nucleator and processive capper that binds to the barbed end of the actin filament [33–35].

Profilin is essential for merozoite invasion of RBCs [36,37] and is required for efficient sporozoite motility [38,39]. Apicomplexan profilin has diverged markedly from higher eukaryotes [36], with low sequence identity (16%) retained between *P. falciparum* profilin (PfPFN)

(PF3D7_0932200) and *Homo sapiens* profilin I (HsPFNI) sequences. This is illustrated by a unique arm-like β-hairpin insertion in apicomplexan profilin that spans from residues 57 to 74 [36]. This unique sequence is essential for actin binding; mutations in this region hinder monomer sequestration in vitro and impact sporozoite motility [38]. Within apicomplexan, PfPFN has also diverged from *Toxoplasma gondii* profilin (TgPFN), with the former acquiring a further acidic loop extension located in residues 40 to 50 [36].

In comparison, actin is more conserved between apicomplexans and higher eukaryotes; however, the apicomplexan actins are among the most diverged actins in eukaryotes. *P. falciparum* encodes 2 actin isoforms: actin-1 (PfACT1) (PF3D7_1246200) and actin-2 (PfACT2) (PF3D7_1412500), the latter of which is only expressed in the sexual stages [40]. PfACT1 has a high degree of sequence identity (93%) with the single actin gene in *T. gondii* [20] and shares an 82% identical sequence with the human cytosolic β actin [14,29,40].

In apicomplexan parasites, actin polymerisation has been difficult to study because of the short length and instability of the actin filaments [14,41,42]. Despite this, actin polymerisation has been shown to be essential in many phases of the lifecycle, including intracellular replication, host cell egress (only in *T. gondii*), motility, and host cell invasion [14,17,43,44]. Of these, host cell invasion is the best-characterised actin-dependent process (reviewed in [44]). In *P. falciparum*, naturally occurring compounds such as cytochalasins D and B, latrunculins, phalloidin, and jasplakinolide have been used to study the complex actin regulation in merozoite invasion [7,29,45–48]. These compounds interfere with actin treadmilling by affecting the polymerisation and depolymerisation of actin through various MoA. For example, cytochalasins bind to the barbed end to prevent polymerisation and can also prevent G-actin disassociation from F-actin [49]; latrunculins prevent sequestration of the G-actin subunits [50]; and phalloidin and jasplakinolide stabilise F-actin by preventing the release of $P_i$ from G-actin-ADP subunits [51]. Apart from the recently developed latrunculins that have greater selectivity [47], these naturally occurring compounds remain biological tools rather than antimalarial candidates due to their cytotoxicity.

Recently, we identified a compound MMV020291 (MMV291) from the Medicines for Malaria Pathogen Box as an inhibitor of *P. falciparum* merozoite invasion of RBCs [52]. We further explored the drug development potential of the compound by defining the structure activity relationship (SAR) and generated analogues with improved potency, while maintaining compound selectivity and invasion blocking activity [53]. Here, through in vitro resistance selection, whole-genome analysis, and reverse genetics, we show that the mechanism of resistance to MMV291 are through mutations in PfPFN and PfACT1. We further explore the MoA of the MMV291, which is the first reported compound series linked to interference with the actin-profilin complex in *P. falciparum*.

## Results

### MMV291-resistant parasites contain mutations in *actin-1* and *profilin*

To select for parasite resistance against our lead molecule MMV291 (Fig 1A), 5 populations of $10^8$ *P. falciparum* 3D7 parasites were exposed to 10 μM (approximately $10 \times EC_{50}$) of the compound until new ring stage parasites were no longer observed by Giemsa-stained blood smears. The drug was removed, and parasites were allowed to recover. This drug on and off selection was performed for 3 cycles before parasite resistance was evaluated in a 72-hour lactate dehydrogenase (LDH) growth assay [54]. This revealed 3 MMV291-selected populations demonstrated an 8- to 14-fold increase in $EC_{50}$ (S1 Fig). These resistant populations (B, C, and D) were cloned out by limiting dilution, and 2 clones from each parent line were tested in an LDH assay, indicating resistance was heritable (Fig 1B). Genomic DNA was extracted from

**A)**

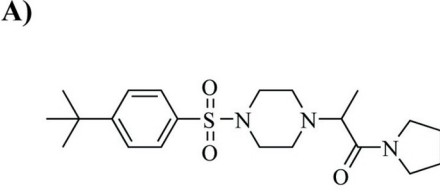

**B) i**

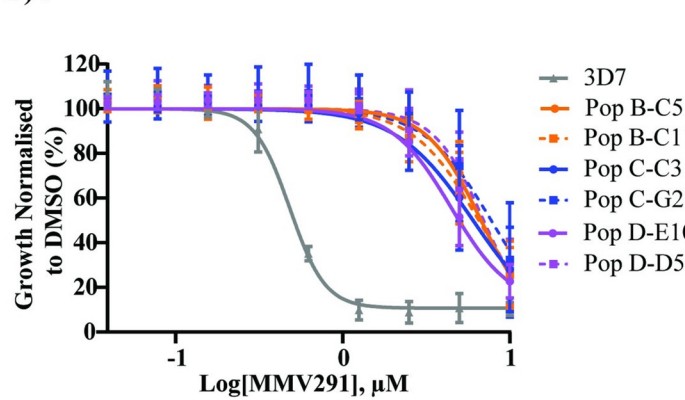

**ii**

| Clone | EC$_{50}$ (μM) |
|---|---|
| **3D7** | 0.48 (0.45–0.57) |
| **Pop B-C5** | 6.74 (4.94–und) |
| **Pop B-C11** | 6.52 (4.67–und) |
| **Pop C-C3** | 6.04 (3.35–und) |
| **Pop C-G2** | 7.75 (3.11–und) |
| **Pop D-E10** | 4.47 (3.19–54.67) |
| **Pop D-D5** | 6.99 (und) |

**C)**

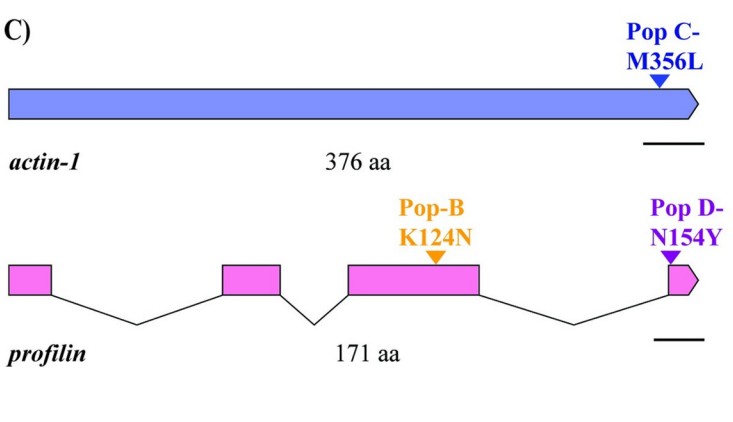

**D)**

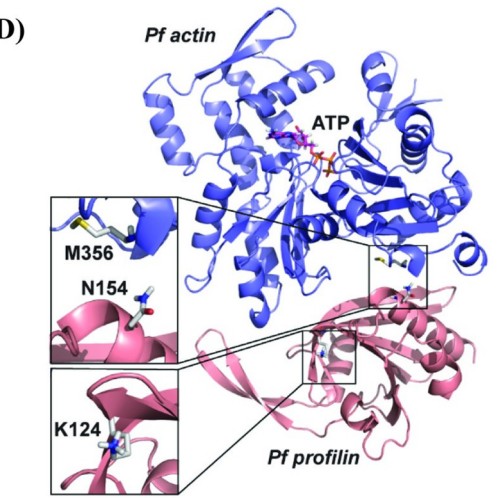

**Fig 1. Resistance selection and whole genome sequencing reveal *actin-1* and *profilin* as candidate proteins involved in the MoA of MMV291. (A)** Chemical structure of MMV291. (**B**) **i** Drug cycling on and off for 3 cycles and subsequent cloning out of parental lines resulted in 2 clones from 3 populations (Pop B, C, and D) that maintained stable resistance to MMV291 in a 72-hour growth assay. Growth has been normalised to that of parasites grown in 0.1% DMSO with error bars representing the standard deviation of 3 biological replicates. **ii** EC$_{50}$ values derived from nonlinear regression curves in GraphPad Prism with 95% confidence intervals shown in brackets. und = undefined. Source data can be found in S1 Data. (**C**) Genome sequencing of the MMV291-resistant parasites revealed a different nonsynonymous single nucleotide polymorphism (SNP) shared by the clonal lines across 2 related proteins; Populations D and B contained K124N and N154Y mutations in profilin (PF3D7_0932200), respectively, while Population C contained a M356L mutation in actin-1 (PF3D7_1246200). Scale bar indicates 100 base pairs. (**D**) The positions of the resistance mutations were mapped onto the X-ray structures of *P. falciparum* actin-1 (purple) (PDB: 6I4E) (42) and *P. falciparum* profilin (pink) (PDB: 2JKG) (36), which revealed PFN(N154Y) and ACT1(M356L) lie on either side of the proteins' binding interfaces. PFN(K124N) resides on the opposing side of profilin. In this case, the X-ray structures of *Oryctolagus cuniculus* actin and human profilin (PDB: 2PBD) (56) were utilised as a template to spatially align the 2 parasite proteins.

these resistant clones, along with a parental 3D7 reference strain, and whole genome sequencing was performed using the Oxford Nanopore MinION platform [55]. Here, a minimum of 10× depth coverage with 70% of the reads to support the called allele was required for verification [55].

Across the 6 clones of MMV291-resistant parasites from 3 populations, there were a total of 18 nonsynonymous single nucleotide polymorphisms (SNPs) identified in 16 genes with no other gene variants found (Table 1). Of these SNPs, 3 were present in related genes across all resistant isolates. One of these SNPs was located in chromosome (Chr) 12:1921849 within the gene encoding *actin-1* (PF3D7_1246200), resulting in an M356L mutation in population C clones. The other 2 SNPs both occurred in the gene encoding *profilin* (PF3D7_0932200), located in Chr 9:1287853 and 1288316, resulting in a K124N and N154Y mutation in population B and D clones, respectively (Fig 1C and Table 1).

A homology model of the binding of *P. falciparum* actin-1 (PfACT1) and profilin (PfPFN) was created using the binding of *Orytolagus cuniculus* actin to *H. sapiens* profilin [36,42,56] (Fig 1D). This indicated that PfACT1(M356) and PfPFN(N154) were located at the binding interface between the 2 proteins, while PfPFN(K124) was orientated away, on the opposite side of PfPFN. Despite the close proximity of these 2 SNPs to the binding site between the 2 proteins, the resistant parasites did not exhibit an associated fitness cost in vitro (S2 Fig), indicating these amino acid changes are well tolerated and may not be essential for actin-1 binding to profilin.

## Mutations in *actin-1* and *profilin* mediate resistance to MMV291 in wild-type parasites

To confirm that the chemically induced PfPFN(N154Y), PfPFN(K124N), and PfACT1 (M356L) mutations were responsible for resistance to MMV291, we employed reverse genetics to introduce each mutation into wild-type (WT) parasites. This was achieved using a CRISPR-Cas9 gene editing system whereby homology regions were designed to both the 5′ and 3′ flanks of the desired loci. The 5′ homology flank encompassed a synthetic recodonised region containing the WT or nonsynonymous drug-resistant mutations and synonymous shield mutations to prevent recleavage with Cas9 after recombination into the desired loci (Fig 2Ai). To direct Cas9 cleavage, a homologous synthetic guide RNA (gRNA) was mixed with a tracrRNA and recombinant Cas9 enzyme and electroporated into blood stage parasites with the donor plasmid [57]. After chromosomal integration was selected with WR99210, viable parasites for both the mutant and WT parasites were confirmed to contain the donor cassette using integration PCRs (Fig 2Aii). These PCR products were sequenced and confirmed to contain the corresponding MMV291-resistant alleles (S3 Fig). Next, the modified lines were tested in an LDH growth assay against MMV291, which showed an 11- to 18-fold increase in $EC_{50}$ in the introduced mutant lines compared to their WT counterparts (Fig 2B). This indicated that PfPFN (K124N), PfPFN(N154Y), and PfACT1(M356L) were responsible for the chemically induced resistance by MMV291, suggesting these proteins are involved in the MoA of the compound.

To date in *P. falciparum*, the dynamics of actin polymerisation have been explored with naturally occurring compounds that bind to various regions within the actin-1 protein [7,45–48,51,58] (S4 Fig). Targeting the actin-binder profilin, however, presents a novel mechanism to interfere with this essential parasite process. This novel MoA of MMV291 was confirmed by the lack of cross-resistance between the chemically induced MMV291-resistant parasites and cytochalasin D (CytD) and jasplakinolide in a 72-hour LDH growth assay (S5 Fig). Furthermore, despite the highly conserved sequence of actin-1 in *H. sapiens* and *P. falciparum* (S6 Fig) and that actin dynamics in RBCs have previously been shown to be perturbed by actin

**Table 1. Resistant genomes pooled variant summary.**

| Chromosome | Position | Ref | Alt | Effect | Gene_ID | Gene_Description | Codon Change | AA_Change | WT_3D7_C9 | WT_3D7 | Pop B_C11 | Pop B_C5 | Pop C_C3 | Pop C_G2 | Pop D_D5 | Pop D_E10 |
|---|---|---|---|---|---|---|---|---|---|---|---|---|---|---|---|---|
| Pf3D7_09_v3 | 1287853 | A | T | nonsynonymous | PF3D7_0932200 | *profilin* | AAA/AAT | K124N | . | . | ALT1 | ALT1 | . | . | . | . |
| Pf3D7_09_v3 | 1288316 | A | T | nonsynonymous | PF3D7_0932200 | *profilin* | AAT/TAT | N154Y | . | . | . | . | ALT1 | . | ALT1 | ALT1 |
| Pf3D7_12_v3 | 1921849 | A | T | nonsynonymous | PF3D7_1246200 | *actin I* | ATG/TTG | M356L | . | . | . | . | ALT1 | ALT1 | . | . |
| Pf3D7_06_v3 | 554517 | T | G | nonsynonymous | PF3D7_0613600 | conserved *Plasmodium* protein, unknown function | TTT/GTT | F21V | . | . | ALT1 | ALT1 | . | . | . | . |
| Pf3D7_03_v3 | 547075 | G | T | nonsynonymous | PF3D7_0313400 | conserved *Plasmodium* protein, unknown function | CAA/AAA | Q370K | . | . | . | . | . | . | . | ALT1 |
| Pf3D7_04_v3 | 682224 | G | A | nonsynonymous | PF3D7_0415300 | cdc2-related protein kinase 3 | GAA/AAA | E139K | . | . | . | . | . | . | . | ALT1 |
| Pf3D7_06_v3 | 524255 | C | T | nonsynonymous | PF3D7_0612600 | cytoplasmic tRNA 2-thiolation protein 1, putative | GGA/GAA | G284E | . | . | . | . | ALT1 | . | . | . |
| Pf3D7_07_v3 | 909422 | C | A | nonsynonymous | PF3D7_0721000 | conserved *Plasmodium* membrane protein, unknown function | CAA/AAA | Q1711K | . | . | . | . | . | . | . | ALT1 |
| Pf3D7_11_v3 | 1266495 | C | T | nonsynonymous | PF3D7_1132600 | pre-mRNA-splicing factor 38A, putative | GAA/AAA | E306K | . | . | . | . | . | . | . | ALT1 |
| Pf3D7_12_v3 | 513882 | C | A | nonsynonymous | PF3D7_1211600 | lysine-specific histone demethylase 1, putative | ACA/AAA | T1187K | . | . | . | . | . | . | . | ALT1 |
| Pf3D7_12_v3 | 1022182 | G | T | nonsynonymous | PF3D7_1225100 | isoleucine—tRNA ligase, putative | GTT/TTT | V1114F | . | . | . | . | . | . | . | ALT1 |
| Pf3D7_12_v3 | 1381681 | C | T | nonsynonymous | PF3D7_1233400 | conserved *Plasmodium* membrane protein, unknown function | AGA/AAA | R233K | . | . | . | . | . | . | . | ALT1 |
| Pf3D7_13_v3 | 1004824 | G | T | nonsynonymous | PF3D7_1324300 | conserved *Plasmodium* membrane protein, unknown function | CCA/CAA | P4827Q | . | . | . | . | ALT1 | . | . | . |
| Pf3D7_13_v3 | 1810063 | G | A | nonsynonymous | PF3D7_1345100 | thioredoxin 2 | GCT/GTT | A84V | . | . | . | . | ALT1 | . | . | . |
| Pf3D7_14_v3 | 271754 | C | T | nonsynonymous | PF3D7_1407600 | conserved *Plasmodium* protein, unknown function | GAA/AAA | E1788K | . | . | . | . | . | . | . | ALT1 |
| Pf3D7_14_v3 | 404507 | G | T | nonsynonymous | PF3D7_1410300 | WD repeat-containing protein, putative | CAA/AAA | Q4186K | . | . | . | . | . | . | . | ALT1 |

(*Continued*)

**Table 1.** (Continued)

| Chromosome | Position | Ref | Alt | Effect | Gene_ID | Gene_Description | Codon Change | AA_Change | WT_3D7_C9 | WT_3D7 | Pop B_C11 | Pop B_C5 | Pop C_C3 | Pop C_G2 | Pop D_D5 | Pop D_E10 |
|---|---|---|---|---|---|---|---|---|---|---|---|---|---|---|---|---|
| Pf3D7_14_v3 | 413594 | C | T | nonsynonymous | PF3D7_1410300 | WD repeat-containing protein, putative | GAA/AAA | E1157K | . | . | . | . | . | . | . | ALT1 |
| Pf3D7_14_v3 | 1464591 | C | A | nonsynonymous | PF3D7_1436100 | conserved Plasmodium membrane protein, unknown function | AAC/AAA | N1068K | . | . | . | . | . | . | . | ALT1 |

Table depicts genes in which nonsynonymous single nucleotide polymorphisms (SNPs) were identified that passed quality filtration and were detected in at least 1 resistant clone. Highlighted rows indicate profilin and actin-1 as the protein candidates involved with the MoA of MMV291.

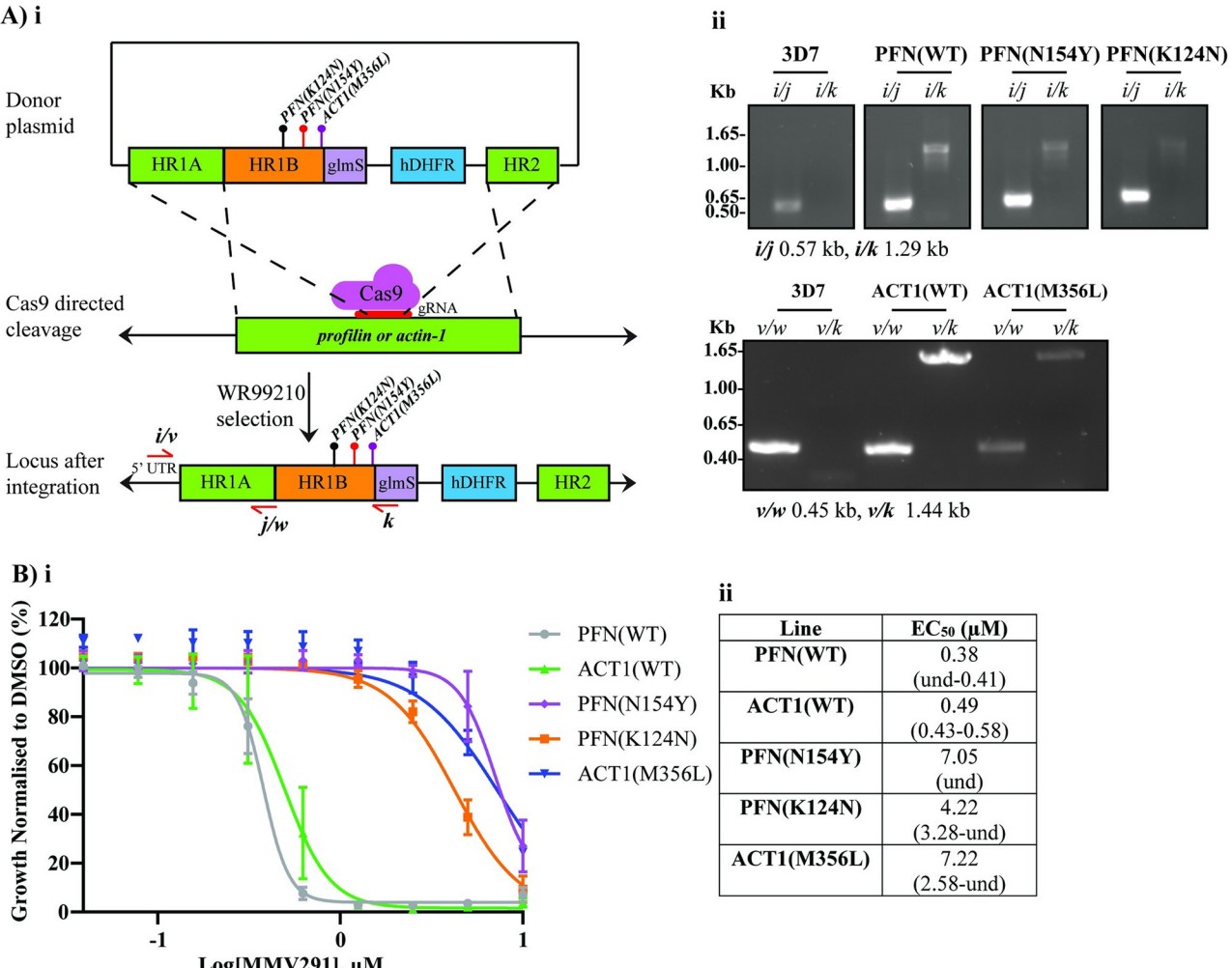

**Fig 2. Introduction of the SNPs in *profilin* and *actin-1* into 3D7 parasites mediates resistance to MMV291. (A) i** Strategy to create the donor plasmid to introduce PFN(N154Y), PFN(K124N), and ACT1(M356L) SNPs into 3D7 parasites. Homology regions (HRs) were designed to the 5′ flank (HR1) and 3′ flank (HR2) whereby HR1 was made up of the endogenous genes' sequence (HR1A) and recodonised fragments (HR1B), encompassing the resistant mutation alleles. A synthetic guide RNA (gRNA) was designed for either *profilin* or *actin-1* to direct Cas9 to the cleavage site and induce double crossover homologous recombination. WR99210 was used to select for integrated parasites via the human hydrofolate reductase (hDHFR). **ii** Integration into the *profilin* or *actin-1* locus was validated whereby a 5′ UTR primer (*i/v*) was used in combination with a primer located in the glmS region (*k*). **B) i** Integrated parasites were tested in a 72-hour LDH growth assay, which revealed the resistant mutations conferred resistance against MMV291 and confirmed the profilin and actin-1 proteins as involved in the MoA of the compound. Growth has been normalised to that of parasites grown in 0.1% DMSO, and error bars indicate the standard deviation of 3 biological replicates. Source data can be found in S1 Data. **ii** EC$_{50}$ values derived from nonlinear regression curves in GraphPad Prism with 95% confidence intervals shown in brackets. und = undefined.

inhibitors [59,60], RBCs that had been pretreated with MMV291 displayed normal levels of merozoite invasion, indicating this compound is not targeting host actin (S7 Fig). Altogether, these data indicate that MMV291 has an alternative MoA from traditional actin polymerisation inhibitors.

## Improving the potency of MMV291 conserves resistance to actin-1 and profilin in mutant parasites

We have previously reported the SAR of MMV291 whereby the alpha-carbonyl *S*-methyl isomer was determined to be important for parasiticidal activity [53]. Four of these analogues, *S*-

W414, *S*-W936, *S*-W415, and *S*-W827 (S8 Fig) (previously referred to as *S*-18, *S*-20, *S*-22, and *S*-38) were selected to study the relationship of the chemical series targeting PfACT1 and PfPFN. These *S*-stereoisomers of the racemic MMV291 compound were tested on 2 clones from each chemically induced MMV291-resistant population in a 72-hour LDH growth assay. This revealed that the resistant clones maintained their resistance against the potent analogues of MMV291 (Fig 3A–D). Interestingly, the degree of resistance differed depending on the parental population; population B clones (PFN(K124N)) were the least resistant, inducing a 10-fold increase in $EC_{50}$ in the 4 analogues, while the population C clones (ACT1(M356L)) exhibited the most resistance, increasing the $EC_{50}$ 60 to 170-fold. These data indicated that since the ACT1(M356L) clones were consistently highly resistant to the MMV291 analogues, the MoA of this chemical series may be linked to PfACT1 function.

## Actin polymerisation is affected by MMV291 in vitro

F-actin detection in apicomplexan parasites has been technically challenging because of the short length of the filaments produced [48,61]. Recently, this has been overcome with the expression of F-actin binding chromobodies in *T. gondii* [43] that have also been adapted to *P. falciparum* [62]. The actin binding chromobodies consist of an F-actin nanobody fused to green fluorescent protein to allow microscopic detection of F-actin, which exists as a distinct punctate signal located at the apical tip of the merozoite. With actin polymerisation inhibitors, such as CytD, the punctate fluorescence dissipates into a uniform signal across the merozoite [62]. To ascertain if MMV291 could inhibit actin polymerisation in merozoites, we treated synchronised schizonts expressing the fluorescent nanobody with the parent MMV291 molecule and 2 analogues; *S*-W936, an active *S*-stereoisomer ($EC_{50}$ of 0.2 μM), and *R*-W936, a less active *R*-stereoisomer of the former molecule ($EC_{50}$ of 6.9 μM) (S8 Fig) at both 5× or 1× the growth $EC_{50}$ (Fig 4). *R*-W936 was also tested at multiples of *S*-W936 $EC_{50}$ to allow a direct comparison between the compound's activity and effect on actin polymerisation. Images of the egressed merozoites were captured and quantification of the punctate versus uniform F-actin signal was scored (Fig 4A). This revealed that at both concentrations of MMV291 and *S*-W936 tested, and high concentrations of less active isomer, *R*-W936, caused a similar reduction in merozoites expressing F-actin puncta to CytD treatment ($P > 0.05$; Fig 4B). In contrast, low concentrations of the less active isomer, *R*-W936, was significantly less effective at preventing merozoites from forming F-actin puncta than CytD ($P < 0.001$; Fig 4B). This result was notable as it provides the first direct link between the parasiticidal activity of MMV291 and its ability to inhibit F-actin formation in merozoites.

Resistance to MMV291 arose due to mutations in both PfACT1 and PfPFN, suggesting the MMV291 series was interacting at the binding interface of the 2 proteins. To dissect the basis of this interaction, in vitro sedimentation assays with recombinant monomeric PfACT1 were carried out in the presence of compounds *S*-MMV291, *R*-MMV291, *S*-W936, *R*-W936, *S*-W414, and *S*-W827 and vehicle control, DMSO. In agreement with previous studies [42,63], 80% of PfACT1 could be sedimented into a pellet fraction by ultracentrifugation in the absence of MMV291 analogues (Figs 5A and S9A), while 15% of PfACT1 could be sedimented in the nonpolymerizing (G-buffer) conditions (S9C and S9D Fig). *S*-W936 was found to cause a small but significant reduction in the amount of PfACT1 in the pellet to 68% ($P = 0.01$; Fig 5A). The remaining compounds had no statistically significant effect on PfACT1 sedimentation. These results indicate that the MMV291 analogues have either no or minimal impact on actin polymerisation in vitro.

To address whether the MMV291 analogues affected the PfPFN–PfACT1 interaction, we included PfPFN in the sedimentation assays. As PfPFN sequesters G-actin, only 21% of

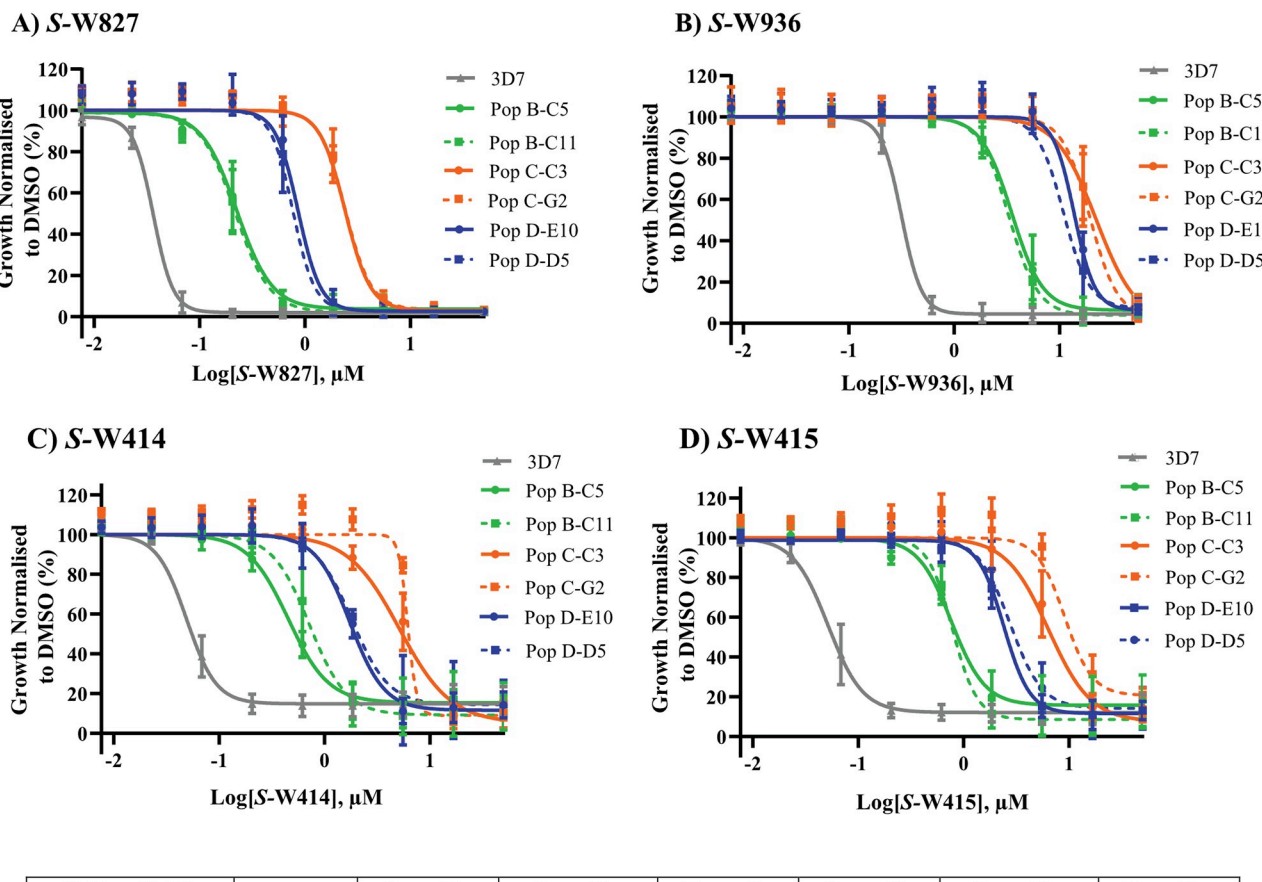

**Fig 3. MMV291-resistant parasites demonstrate varying resistance to 4 analogues of MMV291.** Two clones from 3 independently derived MMV291-resistant parasite lines were tested in 72-hour LDH growth assays. Varying degrees of resistance to *S*-W827 (**A**), *S*-W936 (**B**), *S*-W414 (**C**), and *S*-W415 (**D**) was observed, with Population C clones demonstrating the greatest resistance and Population B clones retaining the most sensitivity to the 4 molecules. Values were normalised to parasite growth in 0.1% DMSO, with error bars representing the mean of 3 biological replicates. Dose response curves were generated in GraphPad Prism using nonlinear regression to derive mean $EC_{50}$ values, which are stated in the table. S.D indicates the standard deviation calculated from $EC_{50}$ values across 3 biological experiments. Heat map indicates degree of resistance from 3D7 control lines, with yellow and red indicating the lowest and highest degree of resistance, respectively. Source data can be found in S1 Data.

PfACT1 remained in the polymerised pellet fraction following sedimentation (Figs 5B and S9B). In the presence of the compounds, the amount of PfACT1 in the pellet decreased significantly between 7.5% to 15% with *S*-MMV291, *R*-MMV291, *S*-W936, *R*-W936, and *S*-W414 treatment ($P < 0.01$; Figs 5B and S9B). *S*-W827 exhibited the greatest affect by decreasing the PfACT1 to approximately 5% ($P < 0.0001$). The magnitude of the effects observed from the

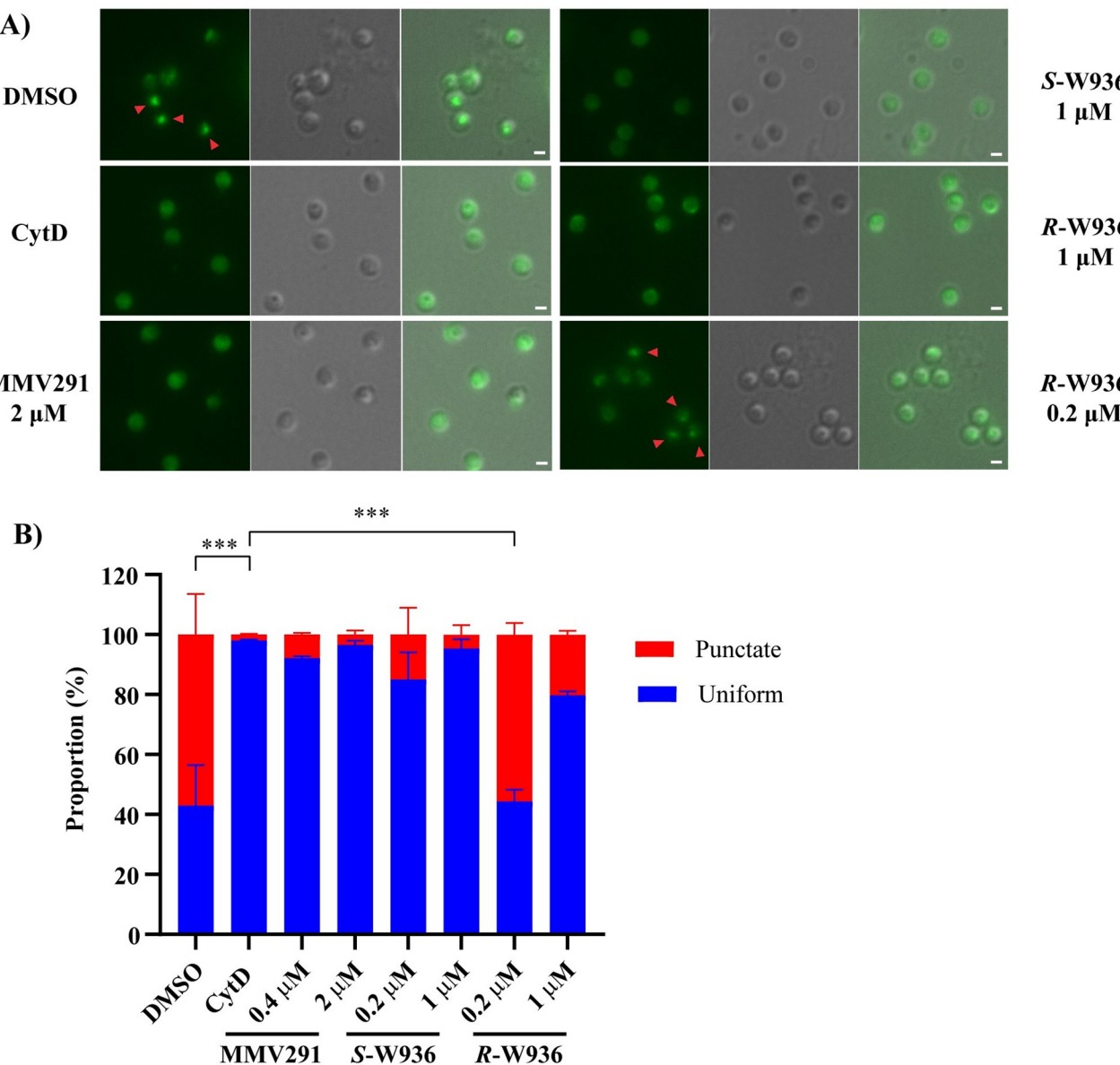

**Fig 4. MMV291 treatment prevents F-actin formation in merozoites. (A)** Synchronised schizonts from a *P. falciparum* parasite line expressing an F-actin-binding chromobody were incubated with DMSO, Cytochalasin D (CytD), MMV291, and analogues *S*-W936 and *R*-W936, for 20 minutes at 37°C to allow merozoite egress. Merozoites were then imaged to detect either a normal punctate apical F-actin fluorescence signal or uniform signal, indicative of the inhibition of F-actin formation. Arrow heads depict punctate F-actin signal, and scale bar indicates 1 μm. **(B)** The proportion of merozoites with a punctate or uniform signal were scored with >550 merozoites counted for each treatment. Merozoites treated with the lower concentrations of the less active *R*-W936 had equal proportions of punctate and uniform fluorescence signals, like the DMSO control. In contrast, CytD, MMV291, and the active *S*-W936 compounds all greatly inhibited the formation of a punctate F-actin signal. Error bars represent the standard deviation of 2 biological replicates, each made up of 3 technical replicates from 3 individual counters. Statistical analysis was performed using a one-way ANOVA, comparing the mean of CytD punctate proportions with the mean of other treatments. *** indicates $P < 0.001$; no bar indicates not significant. DMSO and CytD were used at concentrations of 0.1% and 1 μM, respectively. Source data can be found in S1 Data.

different compounds on actin sedimentation was correlated with the $EC_{50}$ values of the MMV291 analogues (S8 Fig) with the most potent inhibitors of parasite growth causing the greatest reduction in PfACT1 polymerisation. Notably, *R*-MMV291 had the smallest affect in agreeance with the weak parasite activity of this isomer compared to *S*-MMV291. In summary,

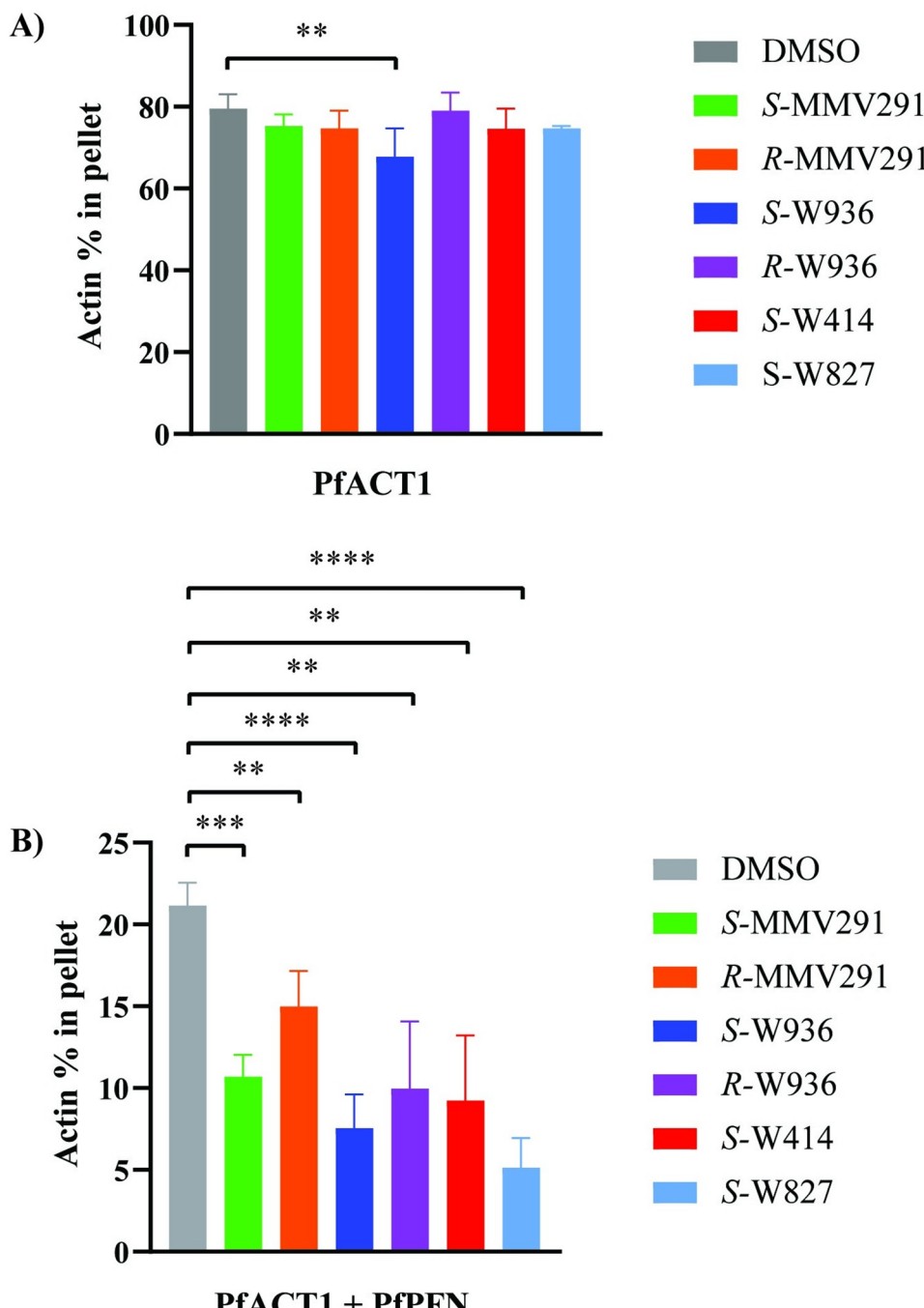

**Fig 5. MMV291 analogues interfere with actin polymerisation in the presence of profilin in vitro.** PfACT1 (4 μM) under polymerizing conditions was quantified in the supernatant and pellet fractions in the presence of the MMV291 analogues (25 μM) or DMSO and upon addition of PfPFN (16 μM). **(A)** In the absence of PfPFN, 80 ± 4% of PfACT1 sedimented to the pellet fraction with the vehicle DMSO treatment. *S*-W936 decreased the amount of actin in the pellet to 68 ± 7%, while the remaining compounds had no significant effects on actin sedimentation. **(B)** Upon addition of PfPFN, actin sedimentation decreased to 21 ± 1% with DMSO treatment. All MMV291 analogues, *S*-MMV291, *R*-MMV291, *S*-W936, *R*-W936, *S*-W414, and *S*-W827, decreased the amount of actin in the pellet further to 11 ± 1%, 15 ± 2%, 8 ± 2%, 10 ± 4%, 9 ± 4%, and 5 ± 2%, respectively. Results are plotted as mean ± standard deviation of the relative amounts of actin in the pellet fraction. The data are based on at least 3 independent assays each performed in triplicate. Statistical significances were determined using an unpaired two-tailed *t* test, where ** $P \leq 0.01$ and *** $P \leq 0.001$, and **** $\leq 0.0001$. No bar indicates not significant. Source data can be found in S1 Data.

these results indicate that the compounds act through a PfPFN-mediated mechanism to interfere with actin polymerisation in parasites.

## MMV291 activity is specific for actin-1-dependent processes in the asexual stage of *P. falciparum* infection

F-actin is required for many processes across the lifecycle of *P. falciparum* including sporozoite gliding motility and hepatocyte invasion [38,64]. However, when sporozoites were treated with MMV291, both of these processes remained unaffected (S10 Fig). Similarly, despite the conserved sequences of actin-1 and profilin in *P. falciparum* and *T. gondii* (S11 Fig), MMV291 and its analogues also had little activity against tachyzoite invasion, unless the compounds were used at high concentrations relative to those used against *P. falciparum* (500 to 1,000 μM) (S12 Fig). Altogether, this, combined with previous data showing MMV291 has little activity against gametocytes [53], indicates that this compound series has activity solely in the asexual stage of *P. falciparum* infection.

Next, we examined the effect of MMV291 on other F-actin-dependent processes in the asexual stage. Conditional knockout of *actin-1* in *P. falciparum* results in a defect in apicoplast segregation [65]. To evaluate the activity of MMV291 against apicoplast segregation, a parasite line that expresses a fluorescently tagged protein destined for trafficking to the apicoplast (ACP-GFP) was utilised [66]. Trophozoites were treated with 5 μM and 10 μM MMV291 (equating to $10 \times EC_{50}$ and $20 \times EC_{50}$) or the vehicle control before being imaged at schizont stages (Fig 6Ai). The schizonts were scored to either have apicoplasts that were reticulated (an immature branched form), segregated (mature form), or clumped (abnormal) [65,66]. This revealed that the DMSO treatment resulted in a majority of normal apicoplast segregation with GFP labelling visualised as distinct punctate signals in daughter merozoites (Fig 6Aii). In contrast, both concentrations of MMV291 induced a defect in apicoplast segregation whereby the 10 μM MMV291 resulted in significantly less segregated apicoplasts than the vehicle control ($P = 0.0003$; Fig 6Aii). This was visualised as distinct "clumps," reminiscent of the phenotype shown previously in *actin-1* knockouts (Fig 6A) [65]. While the 5 μM concentration also displayed less segregation, this number was not significant ($P = 0.18$; Fig 6Aii). Altogether, this indicated that MMV291 induced a dose response effect on apicoplast segregation.

Defects in apicoplast inheritance for daughter merozoites induce a "delayed death phenotype" whereby drugs targeting the apicoplast, such as the antibiotic azithromycin, exhibit no parasiticidal activity until the second cycle of growth after defective merozoites invade new RBCs and progress to trophozoites [67–69]. To investigate if MMV291 also produced a delayed death phenotype, highly synchronous ring-stage parasites expressing an exported nanoluciferase protein were treated with a titration of azithromycin, chloroquine, or MMV291. After 40 hours and prior to merozoite invasion, the compounds were washed out and parasites allowed to grow for a further 2 cycles with nanoluciferase activity used as a marker for parasite growth (Fig 6B). This demonstrated that azithromycin-treated parasites in cycle 1 elicited a dose–response decrease in parasite biomass in cycle three, producing an $EC_{50}$ of 67.5 nM (Fig 6Ci and 6Civ), in contrast to chloroquine, which demonstrated the profile of a fast-acting antimalarial (Fig 6Cii). MMV291 displayed some intermediate delayed death activity at the maximum concentrations tested with cycles 2 and 3 producing $EC_{50}$'s of >10 μM and 7 μM, respectively (Fig 6Ciii and 6Civ). This was significantly higher than the compound's overall $EC_{50}$ of 0.5 to 0.9 μM in a 72-hour LDH assay, suggesting apicoplast segregation and subsequently delayed death is a secondary MoA of MMV291.

While all these data pointed to the MMV291 series having specificity for the asexual stage of *P. falciparum* infection, mostly during merozoite invasion, an outstanding question

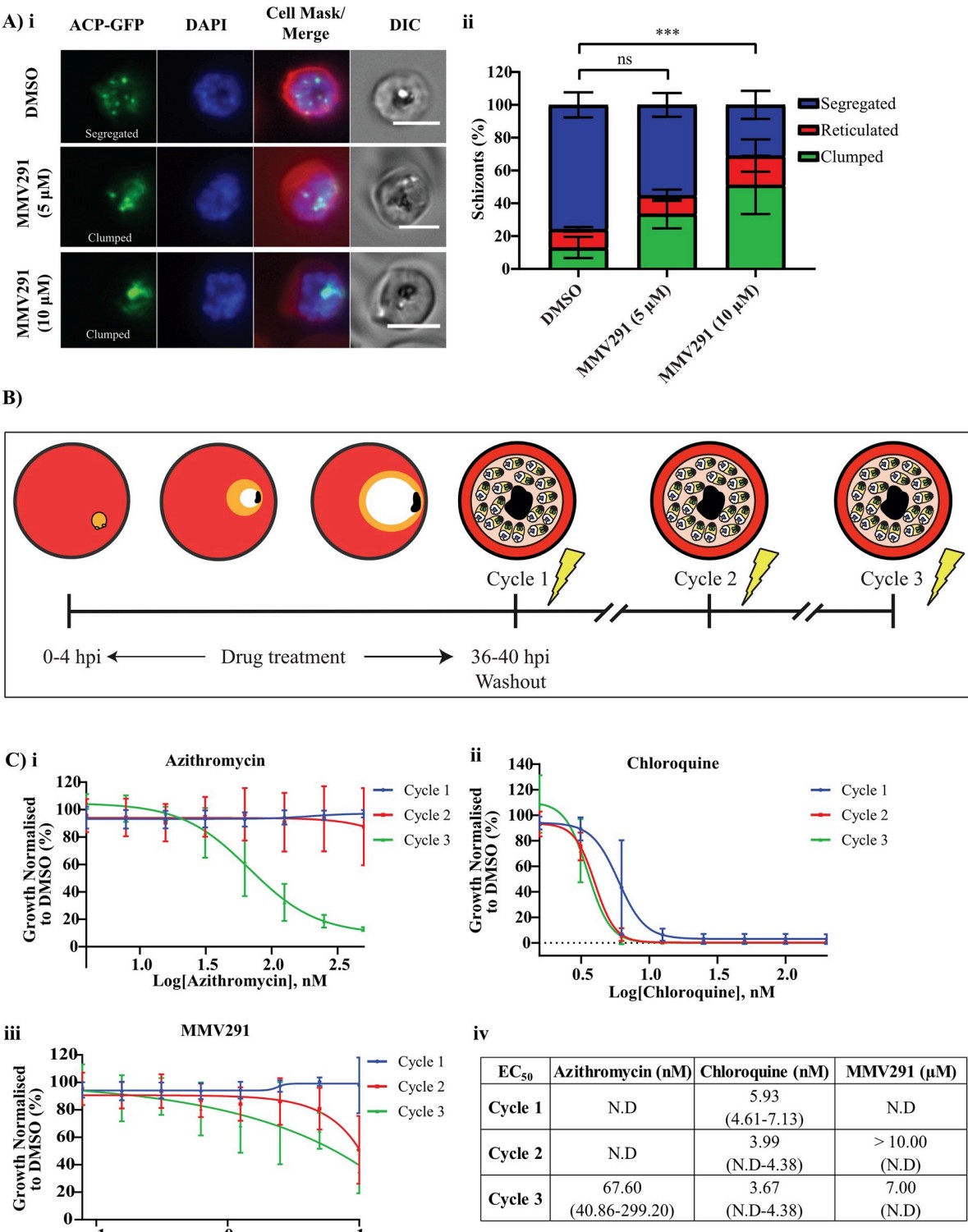

**Fig 6. MMV291 disrupts actin-dependent apicoplast segregation and induces a partial delayed death phenotype. (A) i** Representative panels from live cell imaging of apicoplast targeted acyl carrier protein (ACP) tagged with GFP revealed that treatment of trophozoites for 24 hours with both 5 μM and 10 μM MMV291 (10 and 20 × EC$_{50}$) disrupted apicoplast segregation, resulting in an increase in abnormal apicoplast clumping at schizonts. Scale bar indicates 5 μm. **ii** These images were quantified by 3 independent blind scorers, which showed a significant decrease in the normal segregation of apicoplasts between the 10 μM MMV291 and DMSO treatments, while the 5 μM MMV291

was not significant (ns). The number of cells analysed were 60, 48, and 47 for DMSO, 5 μM MMV291, and 10 μM MMV291, respectively, which were captured over 3 biological replicates. The error bars represent the standard deviation of 3 independent blind scoring. Statistics were performed via a two-way ANOVA using GraphPad Prism between the DMSO segregated panel and the other treatments. *** indicates $P < 0.001$. (**B**) Schematic of the delayed death assay set-up. At 0–4 hours post invasion (hpi) ring-stage parasites expressing nanoluciferase (Nluc) were exposed to titrations of compounds for approximately 40 hours before compounds were washed out. Each cycle for 3 cycles, samples were collected for evaluation of Nluc activity to quantify parasitemia. (**C**) Azithromycin (**i**), chloroquine (**ii**), and MMV291 (**iii**) were evaluated in the delayed death assay where it was found that unlike azithromycin, MMV291 did not display characteristics of a delayed death inhibitor but had partial reduction in parasite growth at the highest concentration used (10 μM) in the second and third cycles. In contrast, the fast-acting antimalarial chloroquine exhibited killing activity in the first cycle. Growth was normalised to that of parasites grown in 0.1% DMSO and EC$_{50}$ values (**iv**) were derived from dose–response curves plotted from nonlinear regressions in GraphPad Prism with 95% confidence intervals of these values specified in brackets. Source data can be found in S1 Data.

important for drug development was whether this chemotype disrupted actin filaments in eukaryotic cells. To address this, we labelled F-actin in HeLa cells and exposed them to the classical actin inhibitors, Latrunculin B and CytD, and increasing concentrations of MMV291 before imaging them by lattice light shield microscopy across 3 hours (S13 Fig). This revealed that the actin inhibitors, Latrunculin B and CytD, had profound effects on disrupting actin filaments even after only 30 minutes of treatment (S13B and S13C Fig and S1 Movie). In comparison, MMV291 treatments (even up to the 20 μM concentration) were seen to have no effect on the filaments, with similar labelling seen to that of the vehicle control, DMSO (S13A and S13D–S13G Fig and S1 Movie). Altogether, this demonstrates that the MMV291 chemotype is specific for disrupting *P. falciparum* actin.

## Discussion

In this study, we sought to uncover the target and explore the MoA of a sulfonylpiperazine, MMV291, which acts to prevent merozoites from deforming and invading human RBCs. Resistance selection coupled with whole genome sequencing revealed 3 independent mutations in PfPFN and PfACT1 that did not impose a fitness cost on parasite growth in vitro. Furthermore, introducing these mutations into WT parasites mediated resistance to MMV291, indicating PfPFN and PfACT1 as proteins involved in the MoA of the compound. Interestingly, the 3 MMV291-resistant populations were observed to produce differing levels of resistance against the more potent MMV291 analogues, with parasites containing the PfACT1(M356L) mutation demonstrating the greatest resistance. This could indicate that MMV291 may interact with higher affinity to PfACT1 and thereby a conservative mutation may lead to reduced MMV291 binding, while still retaining the PfPFN–PfACT1 interaction. In contrast, the other 2 MMV291 PfPFN resistance mutations resulted in more radical amino acid changes and the fact that these mutants elicit similar overall parasite growth as the conservative PfACT1(M356L)-resistant parasites could indicate greater plasticity on the profilin side in PfPFN-PfACT1 binding.

We have previously reported that although MMV291-treated merozoites cannot deform and invade RBCs, the merozoites are still capable of irreversibly attaching to their target RBCs and can subsequently trigger echinocytosis [52]. These characteristics are similar to those reported with CytD, although this naturally occurring compound acts upon the actin filament itself to prevent polymerisation [7,49]. We previously noted that for RBCs with adherent MMV291-treated merozoites that the period of echinocytosis was greatly prolonged and that the adherent merozoites often produced pseudopodial extensions [52]. Both effects have recently been observed in CytD-treated merozoites utilising a lattice light shield microscopy system [70]. Here, membranous protrusions were described projecting from the parasite itself and extending into the interior of the RBC [70]. This CytD defect in merozoite invasion has also been reported as internal "whorls," which were visualised with antibody staining of the rhoptry bulb protein RAP1, indicating that although CytD blocks merozoite entry, rhoptry

release was unaffected [71]. Disruption of RBC integrity due to the injection of merozoite rhoptry contents therefore appears to cause extended RBC echinocytosis unless the merozoite can enter the RBC and reseal the entry pore.

To further investigate the MMV291 series effect on actin polymerisation, in vitro actin sedimentation assays were carried out, revealing the compounds had no activity against PfACT1 polymerisation in the absence of PfPFN, apart from S-W936 that caused a slight reduction. However, all compounds tested significantly enhanced the ability of PfPFN to sequester actin monomers, with the greatest effects observed for the analogues, which most potently inhibited parasite growth. It should be noted that although 2 of these analogues (R-MMV291 and R-W936) have low potency against the RBC stage of P. falciparum (EC$_{50}$ >11 μM and 6.9 μM, respectively), sedimentation assays were carried out at 25 μM, which could explain their activity in PfACT1 sequestration in the recombinant assay. This PfACT1 sequestration effect seen with the MMV291 analogues suggests that this compound series could stabilise the interaction between PfACT1 and PfPFN, leading to decreased actin polymerisation. This could have a profound impact on the formation and turnover of F-actin required for invasion and other cellular functions. Indeed, a downstream effect was observed in parasites expressing an F-actin chromobody whereby the MMV291 series was found to inhibit F-actin in merozoites in a manner that correlated with the parasiticidal activity of the compound. Altogether, this forms the basis of our proposed model of the MoA of MMV291, whereby MMV291 may increase the PfPFN sequestering effect of PfACT1, resulting in less PfACT1 turnover for the formation of the filaments, thereby functionally hindering the actomyosin motor and preventing merozoite invasion of RBCs (Fig 7). While the predictive model of bound PfPFN and PfACT1 places 2 of the 3 resistance mutations in the binding interface between the proteins, the exact binding location and subsequent "target" of MMV291 remains to be uncovered. Further studies into the compounds' effects on the kinetics of actin polymerisation in parasites and crystallography studies solving the binding site of the MMV291 series in relation to the PfPFN-ACT1 interaction would be worthwhile attempting in order to confirm this stabilisation model and gain a greater understanding of the druggable potential of these essential parasite proteins.

The proposed MoA of PfPFN-ACT1 stabilisation within parasites contrasts a previously identified inhibitor of the profilin1/actin interaction in mammalian cells that was found to combat pathological retinal neovascularization [72,73]. Here, the authors confirmed the competitive activity of the compound by demonstrating its inhibition of actin polymerisation in the presence of profilin1 [73], highlighting the druggable potential of this protein–protein interaction. It is thought that apicomplexan profilin may have originated from an evolution fusion of 2 ancestral genes [74], and, therefore, the PfPFN–ACT1 interaction may provide the basis of a selective drug target not found in their mammalian counterparts. This was reinforced by the lack of activity of MMV291 against HepG2 cells [53] or merozoite invasion into RBCs pretreated with MMV291. Additionally, we further extrapolated the selectivity of MMV291 for Plasmodium by confirming that the compound did not affect actin filaments in HeLa cells. Despite the phenotype of MMV291-treated merozoites phenocopying CytD, the MoA of MMV291 interference in actin polymerisation is more reminiscent of the latrunculins. These naturally occurring compounds prevent actin turnover through binding to G-actin subunits [47,50]. While targeting both G-actin and PFN-ACT1 result in a similar phenotype of reduced filament formation, compounds directed at the PfPFN–ACT1 interaction may have more success due to greater selectivity, a phenomenon we observed in our imaging of actin filaments in HeLa cells.

While crucial for merozoite invasion, PfPFN-PfACT1 may not be required for other F-actin-dependent processes such as gametocytogenesis and apicoplast segregation [14,65,75]. The MMV291 series have previously shown little activity against gametocytes [53,76], and while we did observe some activity against apicoplast segregation with MMV291 treatment,

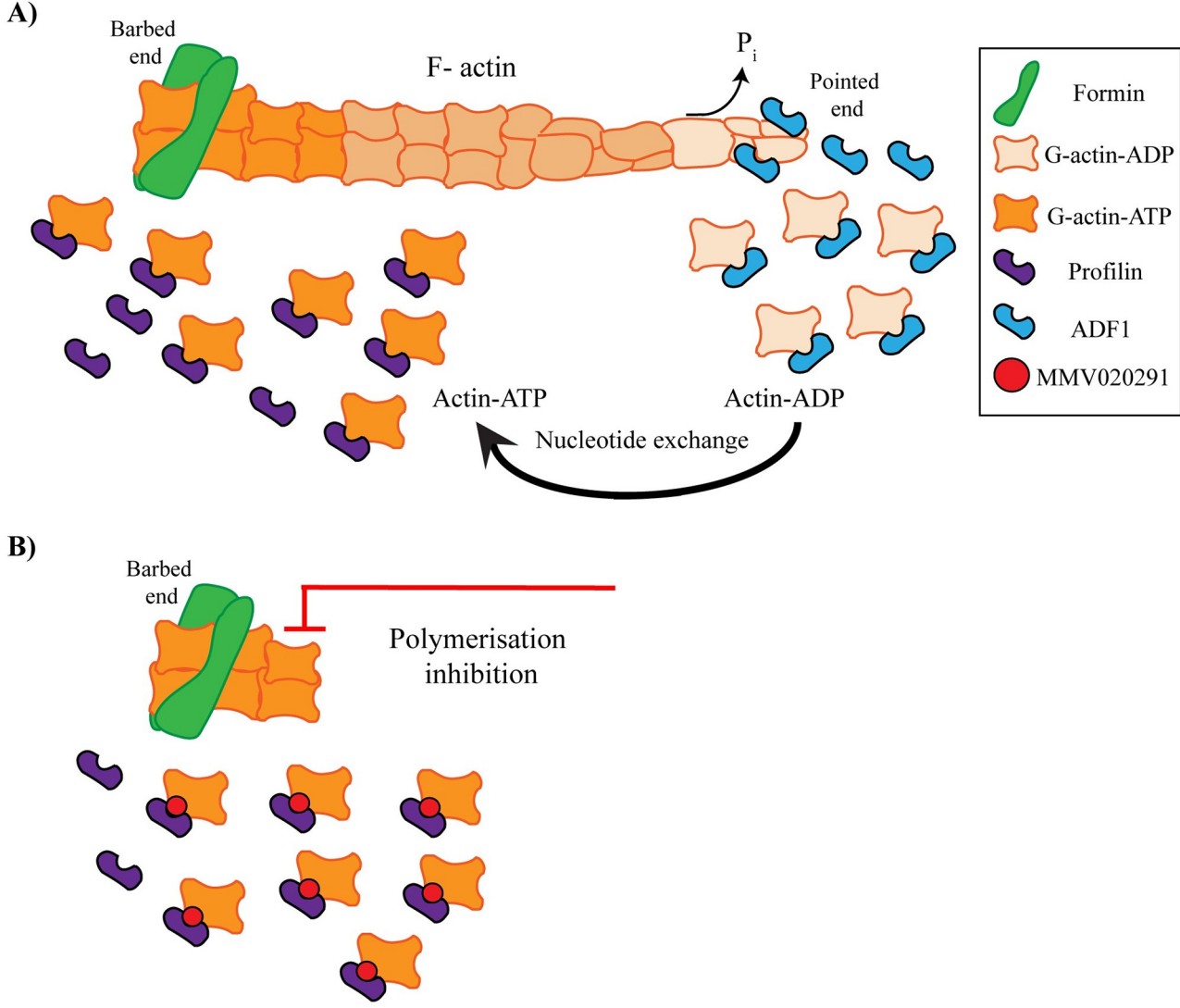

**Fig 7. Proposed model for MMV291 interference in profilin-mediated filamentous actin polymerisation. (A)** Treadmilling model of profilin's role in sequestering G-actin and stimulating the exchange of ADP for ATP before delivering the subunits to the barbed end of the growing filament. Here, formin initiates the polymerisation process to form F-actin. Hydrolysis of the G-actin-ATP occurs at this end to produce G-actin-ADP and inorganic phosphate ($P_i$), to stabilise the filament. The slow release of $P_i$ at the pointed end induces filament instability and proteins such as ADF1 bind to G-actin-ADP to aid in the release of the subunits, thereby severing the filaments. **(B)** A potential mechanism for MMV291's inhibitory activity could be through the stabilisation of the G-actin/profilin dimer therefore inhibiting the formation of F-actin and preventing the generation of force required for invasion. ADF1, actin depolymerising factor 1; F-actin, filamentous actin; G-actin, globular actin.

this parasiticidal activity occurred in much greater concentrations than observed within a standard 72-hour growth assay. This implicates apicoplast segregation as a secondary MoA of MMV291 and perhaps other G-actin sequestering-binding monomers, such as actin-depolymerisation factor 1 (ADF1) [77,78], could be the predominant PfACT1 sequesters that are utilised by parasites for these F-actin-dependent processes. Additionally, the requirements for PfACT1 sequestering and subsequent turnover of F-actin may vary dependent on the process at hand. This can be realised by the varying speeds in motility in different stages of parasites; ookinetes move at 5 μm/min [79–81], merozoites are the next fastest at 36 μm/min [23], while the fastest are sporozoites, which can reach speeds of 60 to 120 μm/min [82,83].

It was somewhat unexpected that MMV291 did not reduce sporozoite motility since sporozoites have been shown to be highly sensitive to mutations in profilin [38,39]. However, these mutations were located in an acidic loop and a conserved β-hairpin domain, which led to the disruption or weakening of the PFN-ACT1 complex and thereby implicating this interaction as an essential requirement for fast-gliding motility [38]. It is therefore possible that our proposed MMV291 MoA of stabilisation of the PFN-ACT1 interaction is not detrimental to actin polymerisation within sporozoites. Furthermore, we showed that hepatocyte invasion of sporozoites were unaffected by MMV291 treatment. This may be due to the different requirements for the PFN–ACT1 interaction to aid in actin polymerisation and subsequent G-actin turnover to invade these host cells with varying membrane tensions and elasticity. Altogether, this indicates this particular interference in the PfACT1–PfPFN interaction appears to specifically inhibit *P. falciparum* invasion of RBCs.

This trend of specificity for merozoite invasion of RBCs was extended to *T. gondii* where tachyzoites also displayed limited sensitivity to the MMV291 series. Here, high concentrations of compounds were required to elicit a reduction host cell invasion. Despite TgPFN being essential for host cell invasion [84] and that TgPFN and PfPFN proteins are somewhat conserved [38], a key difference between these parasites is that TgPFN inhibits the conversion of ADP-ATP on G-actin, thereby inhibiting F-actin polymerisation [85,86]. This has not been observed with PfPFN [14,38]. This highlights the diverged nature of profilin within apicomplexan parasites and, along with differences in host cells, may explain the disparity in activity of the series between *P. falciparum* and *T. gondii*.

Within the *Plasmodium* spp., profilin is highly conserved [36]. MMV291 has previously been shown to possess activity against *Plasmodium knowlesi*, albeit with less potency than *P. falciparum* [53], indicating that there may be a conserved PFN-ACT1 mechanism across *Plasmodium* spp. that is required for invasion of RBCs. Indeed, the resistant mutation locations are conserved in *P. knowlesi* profilin (PkPFN(K125), PkPFN(N155)) but further work as to whether this parasiticidal activity is linked to invasion defects in *P. knowlesi*, and if it extends to other *Plasmodium* spp., is required.

In summary, this investigation identified the first specific inhibitor of *P. falciparum* actin polymerisation in which its MoA could be linked through interference with PfPFN/ACT1 dynamics. The antimalarial development of this compound is currently hampered by the high clearance of MMV291 and its analogues in liver microsomes [53]. Additional medicinal chemistry work is therefore required to address the metabolic instability of this series before it can progress further towards a future antimalarial. Nonetheless, the MMV291 series could serve as a useful tool to study the complex regulation of actin polymerisation in the malaria parasite. This, in turn, could provide a starting point for future development of novel scaffolds against profilin-mediated F-actin polymerisation.

## Materials/methods

### *Plasmodium falciparum* in vitro culturing and parasite lines

*P. falciparum* parasites were cultured as previously reported [87] in human O type RBCs (Australian Red Cross Blood Bank) at 4% haematocrit in RPMI-HEPES supplemented with 10% v/v heat-inactivated human serum (Australian Red Cross) or albumax (Gibco). Unless specified, all assays using *P. falciparum* were conducted on laboratory WT strain 3D7 parasites. An exported Nanoluciferase (Nluc) parasite line (ex-Nluc) [88] was used for Nluc-based assays and episomally maintained using 2.5 nM WR99210 (Jacobus Pharmaceutical Company). For apicoplast segregation assays, a parasite line expressing a green fluorescent protein-tagged acyl carrier protein (ACP-GFP) [66] was used, and this was maintained episomally in parasites by

addition of 0.1 μM pyrimethamine (Sigma Aldrich). A chromobody-emerald fluorescent protein expressing *P. falciparum* parasite line was used for actin-chromobody experiments [62].

## Generation of *P. falciparum* sporozoites

Gelatine-selected *P. falciparum* parasites (NF54e strain, Walter Reed National Military Medical Center, Bethesda) were utilised for gametocyte generation as previously described in [89] with daily media changes. Gametocytes from these cultures were diluted to 0.1% and 0.3% gametocytaemia and fed to *Anopheles stephensi* mosquitoes on artificial membrane feeders. *A. stephensi* mosquitoes (STE2, MRA-128, from BEI Resources) were reared in an Australian Biosecurity (Department of Agriculture and Water Resources)-approved insectary. The conditions were maintained at 27°C and 75% to 80% humidity with a 12-hour light and dark photoperiod in filtered drinking water (Frantelle beverages, Australia) and fed with Sera vipan baby fish food (Sera). The larvae were bred in plastic food trays (cat M612-W, P.O.S.M, Australia) containing 300 larvae, each with regular water changes every 3 days. On ecloding, the adult mosquitoes were transferred to aluminium cages (cat 1450A, BioQuip Products, 2321 Gladwick St. Rancho Dominguez, CA 90220) and kept in a secure incubator (Conviron), in the insectary at the same temperature and humidity and maintained on 10% sucrose. Fed mosquitoes were kept at 27°C in a humidified chamber [90]. Salivary glands of infected mosquitoes (day 21 postinfection) were isolated by dissection and parasites placed into RPMI-1640 media.

## Generation of *Plasmodium berghei* sporozoites

*P. berghei* ANKA WT Cl15cy1 (BEI Resources, NIAID, NIH: MRA-871, contributed by Chris J. Janse and Andrew P. Waters) was used for sporozoite motility assays. *P. berghei* ANKA reporter parasite lines for the in vitro liver stage drug assay PbGFP-Luc con (676m1cl1) RMgm-29 (http://www.pberghei.eu/index.php?rmgm=29) was provided by Andy Waters (University of Glasgow, Glasgow, Scotland) [91].

Animals used for the generation of the sporozoites were 4- to 5-week-old male Swiss Webster mice and were purchased from the Monash Animal Services (Melbourne, Victoria, Australia) and housed at 22 to 25°C on a 12-hour light/dark cycle at the School of Biosciences, The University of Melbourne, Australia. All animal experiments were in accordance with the Prevention of Cruelty to Animals Act 1986, the Prevention of Cruelty to Animals Regulations 2008 and National Health and Medical Research Council (2013) Australian code for the care and use of animals for scientific purposes. These experiments were reviewed and permitted by the Melbourne University Animal Ethics Committee (2015123).

Infections of naïve Swiss mice were carried out by intraperitoneal (IP) inoculation obtained from a donor mouse between first and fourth passages from cryopreserved stock. Parasitemia was monitored by Giemsa smear and exflagellation quantified 3 days postinfection. A volume of 1 μL of tail prick blood was mixed with 100 μL of exflagellation media (RPMI-1640 [Invitrogen] supplemented with 10% v/v foetal bovine serum (FBS) (pH 8.4)), incubated for 15 minutes at 20°C, and exflagellation events per $1 \times 10^4$ RBCs were counted. *A. stephensi* mosquitoes were allowed to feed on anaesthetised mice once the exflagellation rate was assessed between about 12 to 15 exflagellation events per $1 \times 10^4$ RBCs. Salivary glands of infected mosquitoes (days 17 to 24 postinfection) were isolated by dissection and parasites placed into RPMI-1640 media.

## Compounds

Stock concentrations of Azithromycin (20 μg/μL, AK-Scientific), CytD (2 mg/mL, Sigma Aldrich), Latrunculin B (2.5 mM, Tocris), Jasplakinolide (1 mM, Sigma Aldrich), and ML10 (10 mM, Lifearc) were made up in DMSO (Sigma Aldrich). Chloroquine (10 mM, Sigma

 MMV291 prevents red blood cell invasion by the malaria parasite through interference of actin-1/profilin

Aldrich) and Heparin (100 mg/mL, Sigma Aldrich) were dissolved in $H_2O$ and RPMI, respectively. MMV291, S-MMV291, R-MMV291, S-W936, R-W936, S-W414, S-W415, and S-W827 (Walter and Eliza Hall Institute) were dissolved in DMSO to a 10-mM stock solution.

### Resistance selection

Resistant parasites were generated as previously described [92] by exposing a clonal population of $10^8$ 3D7 WT parasites to 10 μM MMV291 (approximately $10 \times 10EC_{50}$). The parasites were incubated with the compounds until the drug-treated parasites began to die off, with the drug replenished daily. The compounds were then removed until healthy parasite replication was observed via Giemsa-stained thin blood smear, upon which compound treatment was resumed. Altogether, the compounds were cycled on and off for 3 cycles until 3 populations of MMV291 were observed to be resistant to the compounds via a growth assay. The resistant lines were cloned out by limiting dilution prior to genomic DNA (gDNA) extraction and their $EC_{50}$ for growth was evaluated following a 72-hour treatment to ensure the resistance phenotype was stable.

### Whole genome sequencing and genome reconstruction

Late-stage parasites from the original 3D7 clonal line and MMV291-resistant clones were harvested via saponin lysis (0.05%, Kodak). A DNeasy Blood and Tissue kit (Qiagen) was then used to extract gDNA from the saponin-lysed pellets following the kit protocol with the exception that additional centrifugation steps were performed to remove hemozoin prior to passing lysates through the DNA binding columns.

Concentration of extracted DNA was evaluated by Qubit Fluorometer (Invitrogen Life Technologies). Average length of the DNA sample was then assessed using Tapestation (Agilent Technologies). Based on concentration and average length of DNA sample, 0.2 pmol DNA was carried forward to prepare sequencing library using 1D genomic sequencing kit SQK-LSK109 and Native barcodes (EXP-NBD104, EXPNBD114) as per manufacturer's instructions (Oxford Nanopore Technologies, UK). For maximum sequencing output, each sequencing run comprised of 3 to 5 samples labelled with distinct Oxford Nanopore native barcodes. A 48-hour sequencing run was performed on a MinION platform with MIN106D Flow cells and MinIT (Software 18.09.1) to generate live fastq files.

After sequencing, fastq files were subjected to demultiplexing and adapter trimming was subsequently performed using Porechop (V0.2.3_seqan2.1.1). Read alignment against the *P. falciparum* 3D7 reference genome was performed using minimap2 (V2.17) default parameters. Alignment files (sam format) were processed with samtools utilities (V1.9) to generate sorted bam files. Genotype likelihoods were then computed using bcftools mpileup (V1.9), configured to skip indels and consider only bases with quality 7 or above. Variant calling was then performed using bcftools multiallelic-caller (V1.9) to generate haploid SNP candidates for each sequenced isolate. Candidate SNPs in blacklisted genomic regions [93] (that is, subtelomeric, centromeric, or internal hypervariable regions) were removed using bedtools subtract (V2.29.2) to retain SNPs in the core genome only. Quality filtration of SNP calls, enforcing a minimum depth of coverage of 10× and requiring at least 70% of reads to support the called allele [55], was performed using custom awk script.

Filtered candidate SNPs for each isolate were then imported into R statistical software (V3.6.1) and overlaps between the 3D7 WT and resistant isolates were examined. To account for differences between the 3D7 reference isolate and our independently cultured 3D7 WT isolate, SNPs present in the 3D7 WT isolate were removed. Functional annotation of non-WT candidate SNPs was performed using the VariantAnnotation package (V1.32.0), retaining only nonsynonymous SNPs. To identify causal resistance variants, biological annotations, including

   

gene ontology terms and expression profiles, were collated for the final set of candidate SNPs using in-house software (PlasmoCavalier).

## Molecular biology and transfection of *P. falciparum*

To introduce the *profilin*-resistant mutations, the donor plasmid was designed whereby the coding sequence of *profilin* (PF3D7_0932200) was recodonised without introns to the bias of *Saccharomyces cerevisiae* and synthesised as gBlock fragments (Integrated DNA Technologies) for both the WT and N154Y sequences. The cloning strategy involved amplifying the first 684 bp of the 5′ native sequence of *profilin* using gDNA to form homology region 1A (primers listed in S1 Table, a/b) and the 3′ recodonised gBlock fragment comprising of the last 292 bp of either the WT or N154Y *profilin* to form homology region 1B (primers listed in S1 Table, c/d). The 2 products were sewn together using primers listed in S1 Table (a/d) to reconstitute the full coding sequence of *profilin* (HR1). The 3′ flank of the native *profilin* sequence was amplified using primers stated in S1 Table (e/f) to create the 3′ homology region 2 (HR2). To introduce the K124N mutation, primers outlined in S1 Table (g/h) were used to generate a HR1 (K124N) using the WT sequence as a template. Both HR1(WT/N154Y/K124N) and HR2 were digested with *Bgl*II/*Spe*I or *Eco*RI/*Kas*I, respectively, and inserted into the parasite vector p1.2 [94].

To introduce the *actin-1*-resistant mutation, *actin-1* (PF3D7_1246200) was recodonised to a *Spodoptera frugiperda* bias and synthesised as a gBlock fragment (Integrated DNA Technologies). To create the full length HR1, the first 517 bp of the 5′ end of *actin-1* (HR1A) and 614 bp of the 3′ recodonised gBlock (HR1B) were amplified using gDNA and the gBlock fragment as templates, respectively (primers in S1 Table l/m, n/o). The 2 fragments were then sewn together using primers listed in S1 Table (l/o) to reconstitute the coding sequence. The M356L mutation was introduced using overlapping primers as stated in S1 Table (r/s). The HR2 sequence of *actin-1* was amplified from gDNA using the primers listed in S1 Table (p/q). An internal *Bgl*II site in the HR2 of the native *actin-1* sequence was repaired using overlapping primers in S1 Table (t/u). Both HR1(WT/M356L) and HR2 were introduced into parasite vector p1.2 as outlined above.

The gRNA sequences were designed with a protospacer adjacent motif (PAM) site using the online program https://chopchop.cbu.uib.no/ [95] (sequences in S1 Table). The donor constructs were linearized and transfected into WT 3D7 ring-stage parasites along with recombinant Cas9 enzyme and annealed gRNA and tracrRNA (Integrated DNA Technologies) as described in [57]. Chromosomal integration of the construct, which includes the human dihydrofolate resistance gene (hDHFR), was selected for with 2.5 nM WR99210. Once viable parasites were obtained, gDNA was extracted and integration PCRs were performed with expected products for modified and parental loci (primers listed in S1 Table). These PCR products were sequenced (Micromon Sanger sequencing) to confirm presence of resistant alleles.

## Delayed death assay

The assay was adapted from a previously described method [69] and utilised ex-Nluc parasites that had been tightly synchronised using 25 nM ML10. Ring-staged parasites (0 to 4 hours postinvasion) were diluted in 96-well U-bottom plates to 1%, 0.1%, and 0.01% parasitemia for cycle one, cycle two, and cycle three, respectively, each with 3 technical replicates. The parasites were then distributed into serially diluted MMV291, azithromycin, and chloroquine from 10 μM, 0.5 μM, and 0.2 μM, respectively, in a 2-step dilution. Parasites were incubated with the compounds for approximately 40 hours until they reached the late-trophozoite to

early-schizogony stage and cycle 1 plates were frozen. The remaining plates were washed 3 times before pellets were resuspended to a final volume of 100 μL. The plates were then incubated at 37˚C for a further 48 hours before cycle 2 plates were frozen. Cycle 3 plates were grown for a further 48 hours before also being frozen. At the completion of the assay, plates were thawed and the Nluc signal was quantified by adding 5 μL of whole iRBCs to 45 μL of 1 × Nanoglo Lysis buffer with 1:1,000 NanoGlo substrate (Promega) in a white luminometer 96-well plate. A CLARIOstar luminometer (BMG Labtech) was used to measure relative light units (RLUs) and growth was normalised to 0.1% DMSO.

## Pretreatment RBCs assay

Fresh RBCs (2 μL of 50% haematocrit) were added to either 100 μL of 0.1% DMSO, 10 μM MMV291, 100 μg/mL heparin, or 0.0025% glutaraldehyde (ProSciTech) in a 96-well U-bottom plate and incubated for 30 minutes at 37˚C, with shaking at 400 rpm. RBCs were then washed 3 times and resuspended in a final volume of 50 μL. Invasion assays were then carried out as previously described in [9,52,96,97], whereby purified merozoites expressing an ex-Nluc were then added to pretreated RBCs and naïve RBCs (i.e., RBCs that had not been pretreated with compounds). Invasion efficiency was then evaluated as described in [52].

## Lactate dehydrogenase (LDH) Malstat growth assay

These were performed as previously described in [54]. Synchronous ring-staged parasites were diluted to 0.3% parasitemia and added to serial dilutions of compounds in a 96-well U-bottom plate, with a final haematocrit and volume of 2% and 100 μL, respectively. Plates were incubated for 72 hours at 37˚C and were then frozen at −80˚C until assayed. After thawing, 35 μL of parasite culture was added to 75 μL of Malstat reagent in a 96-well flat-bottom plate and incubated in the dark for 30 to 60 minutes until colour change occurred. Absorbance (650 nm) was measured on a Multiskan Go plate reader (Thermo Scientific), using Skan IT software 3.2. Growth was then expressed as a percentage of vehicle control and dose–response curves plotted using GraphPad Prism 8.4.0.

For the multicycle growth assays, the parasitemia of ring-stage MMV291-resistant clones, E10, B11 and C3, and 3D7 parasites were counted and adjusted to 0.3% parasitemia with 2% haematocrit. Over 10 cell cycles, samples were taken at each cycle and frozen until completion of the assay. To ensure overgrowth of parasites did not occur, at each cycle, parasites were diluted 1 in 8, which was accounted for in the analysis. Growth was measured by an LDH growth assay as outlined above.

## Actin-binding chromobody assay

A *P. falciparum* 3D7 parasite strain expressing a chromobody-emerald fluorescent protein [62] was synchronised using Percoll (Sigma Aldrich) purification and sorbitol lysis and grown for 45 hours to schizont stages. Compound 2 (2 nM) was administered to schizonts and incubated for 3 hours at 37˚C. The inhibitor was then washed out and schizonts were returned to prewarmed complete RPMI media containing either MMV291, *S*-936, *R*-936, CytD, or DMSO and added into a microscope chamber. Here, schizonts were incubated at 37˚C for 20 minutes to allow merozoite egress before live imaging of newly egressed merozoites were conducted. Quantification of images was conducted by 3 independent scorers. Parasites were imaged using the Leica DMi8 widefield microscope attached to a Leica DFC9000 GTC camera, using a 100× objective.

## Apicoplast segregation assay

ACP-GFP parasites [66] were treated at the trophozoite stage with 5 or 10 μM MMV291 or the vehicle control for 24 hours until they reached the schizogony stage. Prior to imaging, they were incubated with 5 μg/mL CellMask DeepRed (Thermo Fisher Scientific) and 0.3 μM of 4′,6-diamidino-2-phenylindole (DAPI) in RPMI with decreased albumax (0.13%) in complete RPMI for 30 minutes. The cells were then washed 3 times in complete RPMI, mounted and imaged on a Zeiss Cell Observer widefield fluorescent microscope. Apicoplasts were scored by 3 independent blinded scorers as fully segregated, reticulated (branched), or clumped (not segregated). Statistical tests were performed in GraphPad Prism 8.4.0 using a two-way ANOVA with multiple comparisons between each treatment group.

## Sporozoite motility assay

Eight-well chamber slides (PEZGS0816 Millipore Millicell EZ SLIDE) were coated with a monoclonal antibody (mAb) specific for the repeat region of the circumsporozoite protein 3D11 mouse anti-PbCSP (RRID:AB_2650479) [98] at a 1 in 1,000 dilution in phosphate buffered saline (PBS) for 30 minutes at 37˚C. Approximately 8,000 to 10,000 sporozoites were incubated in 100 μL of RPMI 1640-HEPES Glutamax (Invitrogen) supplemented with 10% heat-inactivated FBS (Invitrogen), with the appropriate drug concentrations and solvent controls for 30 minutes at 22˚C. The sporozoites were then seeded into a coated well per treatment and allowed to glide for 45 minutes at 37˚C in a 5% $CO_2$ incubator. Experiments were stopped by removing the supernatant and fixing with 4% v/v paraformaldehyde in 1× PBS at 37˚C for 15 to 20 minutes. Primary antibody of PbCSP (courtesy of S. Tan) or PfCSP (courtesy of J. Boddey) (1/1,000 dilution in 3% bovine serum albumin (BSA) in 1×PBS) was applied for 45 minutes followed by a secondary antibody goat anti-mouse IgG, Alexa Fluor R 488; (Invitrogen) (1/1,000 dilution in 3% BSA in 1× PBS). After each of these stages, the wells were washed gently with 1× PBS. Finally, Hoechst 33342 (Sigma Aldrich) in 1× dPBS was added to each of the wells at a final concentration of 5 μg/mL, incubated for 5 minutes, washed with $dH_2O$, and air dried. The number of sporozoites with and without trails was counted under a fluorescent microscope (Olympus CKX41) after 10 μL of DAKO (Sigma Aldrich) and a coverslip were applied.

## In vitro liver stage assay

This was performed as described in [91] but with the following variations. In vitro human liver HCO4 cells (ATCC) were seeded at $1 \times 10^5$ cells/mL, 200 μL in each well in a 96-well Nunc Edge 96-Well, Nunclon Delta-Treated, Flat-Bottom Microplate (Thermo), and grown for 24 hours in Advanced MEM (Invitrogen), 10% (vol/vol) FBS (Invitrogen), and 1% (vol/vol) antibiotic–antimycotic (Invitrogen) in a standard tissue culture incubator (37˚C, 5% $CO_2$). A total of 20,000 sporozoites from freshly dissected infected mosquitoes were added per well. After 52 hours, the supernatant was removed gently, the wells washed with 1× PBS, and after the removal of the PBS, the plate was placed in the −80˚C freezer for at least 30 minutes. To perform the luciferase assay, the plate was removed from the −80˚C and 20 μL of 1× cell culture lysis reagent (CCLR Promega, cat. no. E1531) was added to the frozen plate. The plate was shaken at room temperature for 15 to 20 minutes. This lysate was transferred to Nunc Micro-Well 96-Well, Nunclon Delta-Treated, Flat-Bottom Microplate (cat: 236105 Thermo Scientific). Luciferase assay substrate solution (20 μL, Luciferase Assay System Kit Promega, cat. no. E1500) was added into each of the wells of the lysed samples. RLU for each sample was then measured via a micro plate reader (EnSpire Perkin Elmer).

## *Toxoplasma gondii* invasion assays

Freshly egressed Nluc expressing parasites were harvested and passed through a 25-gauge needle 3 times to liberate from host cells. Parasites were counted and then preincubated for 20 minutes with different concentration of the compounds, ranging from 1,000 μM to 8 μM diluted in DMEM supplemented with 5% FBS. Parasites were then transferred into 96-well plates containing human foreskin fibroblasts in triplicate and centrifuged at 290$g$ for 5 minutes at room temperature. Plates were then transferred to an incubator and the parasites were allowed to invade for 1 hour at 37˚C in the presence of 10% $CO_2$. Wells were then washed with DMEM 4 times to remove any noninvaded parasites. Plates were then placed at 37˚C with 10% $CO_2$ for 24 hours in 200 μL DMEM supplemented with 5% FBS. Invasion media was then removed and the host cells containing the Nluc expressing parasites were then lysed using Promega Nano-Glo luciferase assay kit and the light units quantified on a Millennium Science plate reader.

## Recombinant protein production and actin sedimentation assays

WT PfPFN cloned into pET28a(+)-TEV using NdeI/BamHI cloning site was ordered from GenScript (Leiden, the Netherlands), expressed *in E. coli* BL21(DE3) cells, and purified using standard protocols as described in [39]. As an exception, the purification tag was cleaved with TEV during dialysis. PfACT1 was produced in *S. frugiperda* Sf21 cells (Invitrogen) as described in [99], with a few changes in the protocol. When infecting the cells, 13.5 μL of high-titer virus was used per $10^6$ cells, the cells were harvested 4 days after infection and used right away for protein purification as described in [38].

Sedimentation of 4 μM PfACT1 1 in 10 mM HEPES (pH 7.5), 0.2 mM $CaCl_2$, 0.5 mM ATP, 0.5 mM TCEP, and 2.5% DMSO was studied alone and in the presence of 25 μM MMV291 analogues or in the presence of 16 μM PfPFN without and with 25 μM MMV291 analogues. Actin polymerization was induced by adding polymerizing buffer to final concentrations of 50 mM KCl, 4 mM $MgCl_2$, and 1 mM EGTA. For control purposes, PfACT1 samples without polymerizing buffer were included to the assay. Total sample volume was 150 μL. Samples were polymerized overnight (approximately 16 hours) at room temperature (approximately 22˚C), 100 μL of each sample (in triplicate) were centrifuged for 1 hour at 20˚C using 100,000 rpm and TLA-100 rotor (Beckman Coulter, CA, USA). The resulting supernatants and pellets were separated, the supernatants were mixed with 25 μL of 5× SDS-PAGE sample buffer (250 mM Tris–HCl (pH 6.8), 10% SDS, 50% glycerol, 0.02% Bromophenol Blue, and 1.43 M β-mercaptoethanol), and the pellets were suspended in 125 μL of 10 mM HEPES (pH 7.5), 0.2 mM $CaCl_2$, 0.5 mM ATP, 0.5 mM TCEP supplemented with 1× SDS-PAGE sample buffer. Samples were incubated for 5 minutes at 95˚C, and then 10 μL of each sample was analysed on 4% to 20% Mini-PROTEAN TGX gel (Bio-Rad Laboratories, CA, USA). The protein bands were visualized with PageBlue stain (Thermo Scientific, MA, USA). Gels were imaged using the ChemiDoc XR S+ system (Bio-Rad), and protein band intensities were determined with the ImageJ 1.52D software [100]. For each supernatant and pellet pair, the relative amounts of PfACT1 in pellets were presented as percentages of the total intensity of PfACT1, which was set to 100%.

## Lattice light-sheet microscopy of actin filaments in HeLa cells

HeLa cells (6 × $10^4$) were seeded onto μ-Slide 8 Well Glass Bottom chambers (IBIDI GMBH, 80827) in culture medium (DHEM supplemented with 1% Glutamax and 10% FBS) and incubated at 37˚C in 5% $CO_2$ for 2 days prior to filming. The cells were stained for 1 hour before imaging in 1 μM of SiR-actin (Spirochrome) diluted in culture medium. Directly before

imaging, the compounds were then added to final concentrations of 20, 10, 5, and 2.5 μM for MMV291, 200 nM for CytD, and 5 μM for Lactrunculin B in 50 μL of culture medium.

Imaging was performed using the Zeiss Lattice Light Sheet 7 (LLS7, Zeiss—Pre-serial). Time-lapse imaging was acquired using light sheets (640 nm) of 30 μm length with a thickness of 700 nm created at the sample plane via a 13.3 × 0.44 NA objective. Fluorescence emission was collected via a 44.83 ×, 1 NA detection objective. Aberration correction was set to a value of 182 to minimise aberrations as determined by imaging the Point Spread Function using 100 nm fluorescent microspheres at the coverslip of a glass bottom chamber slide. Data were collected with a frame rate of 20 ms and a y-step interval of 300 nm. Data were collected immediately following treatment (MMV291, CytD, Latrunculin B) addition at a rate of 1 volume every 5 minutes for 3 hours. A volume of 250 μm × 250 μm × 30 μm was acquired for each time point. All conditions were imaged in parallel across multiple wells of the 8-well chamber slide. Light was collected via a multiband stop, LBF 405/488/561/633, filter. Data are presented as a Maximum Intensity Projection (MIP) with contrast adjusted and scaled from 100 to 400 counts for visualisation purposes.

## Supporting information

**S1 Fig. In vitro resistance to MMV291.** Viable parasites recovered after 3 rounds of drug cycling were tested against a titration of MMV291 in a 72-hour lactate dehydrogenase (LDH) growth assay. Parasite growth was normalised to parasite's grown in 0.1% DMSO, which indicated 3 resistant populations were obtained (B, C, and D) with an 8- to 14-fold increase in $EC_{50}$ compared to 3D7. Data points represent the average of 3 technical replicates. C.I indicates 95% confidence intervals for $EC_{50}$ values, which were derived from nonlinear regression curves in GraphPad Prism. N.D = not determined. Source data can be found in S1 Data.
(TIF)

**S2 Fig. MMV291-resistant lines do not have reduced parasite fitness.** The growth of 3 MMV291-resistant population clones, Pop D-E10, Pop B-C11, and Pop C-C3, with the corresponding PFN(N154Y), PFN(K124N), and ACT1(M356L) mutations, along with 3D7 WT parasites, were compared in a 10-cycle growth assay. During each cycle, an aliquot of culture was harvested from each parasite line and frozen until completion of the assay, whereby parasite lactate dehydrogenase was measured as a marker for parasite growth. The fold change in parasitemia was calculated from the previous cycle for each parasite line, which was then expressed as a percentage of the 3D7 fold change. This demonstrated that there was no comparative growth defect associated with the resistant lines, indicating that the mutations in profilin and actin-1 did not reduce the fitness of these parasites. Error bars represents the standard deviation from 1 experiment comprising of 3 technical replicates. Source data can be found in S1 Data.
(TIF)

**S3 Fig. Chromatograms from integrated parasites containing the MMV291-resistant alleles.** The products produced from diagnostic PCRs were sequenced, and the resistant mutations were confirmed to be present for **(A)** K124N (AAG-AAT) and N154Y (AAC-TAC) in *profilin* and **(B)** M356L (ATG-TTG) in *actin-1*.
(TIF)

**S4 Fig. Model of *P. falciparum* profilin and actin-1 with known actin binders.** *P. falciparum* profilin (pink) (PDB: 2JKG) ([36]; structure 16: 1638) and actin-1 (blue) (ATP, magenta) (PDB: 6I4E) [42] heterodimeric complex showing regions of the proteins where actin inhibitors are known to bind relative to the MMV291 *P. falciparum* mutations. The actin inhibitors

aligned to *P. falciparum* actin-1 and shown are Bistramide A (blue) (aligned from *O. cuniculus* actin, PDB: 2FXU) [101], Cytochalasin D (grey) (aligned from *D. melanogaster* actin, PDB: 3EKU) [102], Jasplakinolide (yellow) (aligned from *P. falciparum* F-actin, PDB: 5OGW) [48], Latrunculin B (gold) and Pectenotoxin-2 (green) (aligned from *O. cuniculus* actin, PDB: 2Q0U) [103], Phalloidin (maroon) (aligned from *G. gallus* F-actin, PDB: 7BTI) [104], and Rei-dispongiolide A (orange) (aligned from *O. cuniculus* actin PDB: 2ASM) [105].
(TIF)

**S5 Fig. MMV291-resistant lines are not cross-resistant to other actin polymerisation inhibitors.** A titration of the actin polymerisation inhibitor, Cytochalasin D (CytD) **(A)**, and actin polymerisation stabiliser, Jasplakinolide **(B)**, were tested against the MMV291-resistant lines and 3D7 parasites in a 72-hour growth assay. This revealed that MMV291-resistant parasites did not exhibit cross resistance to CytD and Jasplakinolide, indicating that MMV291 has an alternate mechanism of action. Growth has been normalised to that of parasites grown in 0.1% DMSO, and error bars represent the standard deviation of 3 biological replicates. $EC_{50}$ values were derived from nonlinear regression curves in GraphPad Prism with 95% confidence intervals of these values specified in brackets. Source data can be found in S1 Data.
(TIF)

**S6 Fig. A model of the comparison between mutation locations in human and *P. falciparum*.** An X-ray structure human profilin (gold) (PDB: 2PBD) and a homology model of human actin (green) (created by SWISS-MODEL [106] using *O. cuniculus* actin (PDB: 2PBD) [56] aligned with *P. falciparum* profilin (pink) (PDB: 2JKG) [36] and actin (blue) (PDB: 6I4E) [42] showing the similarity of the heterodimeric complex. The positions of the MMV291 *P. falciparum* mutations and the associated human amino acids are shown for comparison.
(TIF)

**S7 Fig. MMV291 pretreatment of uninfected RBCs does not inhibit merozoite invasion.** Uninfected RBCs were pretreated with invasion inhibitory compounds (blue); 10 μM MMV291, 100 μg/mL heparin, 2 μM cytochalasin D (CytD), or 0.0025% glutaraldehyde (Glut) for 30 minutes at 37˚C, after which the cells were washed (W/O) to remove the inhibitors. Purified merozoites were then allowed to invade the pretreated RBCs. In parallel, merozoites were added to untreated RBCs in the presence of these inhibitors (red). After incubation for 30 minutes at 37˚C, the compounds were washed out and parasites allowed to grow for 24 hours. Successful invasion was assessed by measuring the bioluminescence levels of trophozoite-stage parasites expressing a nanoluciferase reporter, and invasion rate was normalised to the DMSO vehicle control. This demonstrated that unlike the fixative glutaraldehyde, pretreatment with MMV291 did not reduce merozoite invasion of RBCs, producing a similar profile to the invasion inhibitory molecules, heparin and CytD. Error bars represent the standard deviation of 2 biological replicates with statistical analyses performed in GraphPad Prism using a one-way ANOVA with pretreated RBCs compared to glutaraldehyde (blue) and merozoite treatment compared to heparin (red). **** indicates $P < 0.0001$; ns indicates not significant ($P > 0.05$). Source data can be found in S1 Data.
(TIF)

**S8 Fig. MMV291 analogue structures.** The chemical structures and corresponding $EC_{50}$ values against the RBC stage of *P. falciparum* used in this study with original compound names from Nguyen and colleagues (2021) [53] specified in brackets.
(TIF)

**S9 Fig. Actin sedimentation assay gels and quantification of the actin G-buffer control.**
Sedimentation samples consisting of 4 μM PfACT1 (**A**) and with 16 μM PfPFN (**B**) in the presence of 25 μM MMV291 analogues or DMSO as well as 4 μM PfACT1 alone in G-buffer (**C**) were analysed on 4%–20% Mini-PROTEAN TGX gels and visualized with PageBlue stain. S denotes supernatant and P pellet. **D**) Quantification of the relative amount of actin in the pellet fraction for PfACT1 in G-buffer. 15 ± 9% of PfACT1 sedimented to the pellet fraction in G-buffer. Results are reported as mean ± standard deviation. Source data can be found in S1 Data. The data are based on at least 3 independent assays each performed in triplicate, with a representative gel presented.
(TIF)

**S10 Fig. MMV291 has no effect on sporozoite motility or invasion.** (**A**) Sporozoites expressing GFP were used to measure motility via the quantification of fluorescent trails. This revealed that similarly to DMSO, MMV291 had no effect on sporozoite motility in *P. berghei* (**i**) or *P. falciparum* (**ii**), while cytochalasin D (CytD) significantly reduced motility. 3+ indicates 3 or more trails observed. (**B**) In vitro human liver cells were incubated with a titration of MMV291 in the presence of 20,000 sporozoites expressing a luciferase protein. After 52 hours, cells were lysed and luciferase activity was measured to correlate with sporozoite invasion rate. In contrast with CytD (10 μM) treatment, MMV291 did not reduce invasion rate of sporozoites at concentrations tested. Error bars represent the standard deviation across 3 biological replicates each comprised of 3 technical replicates. Statistical analysis performed via a chi-squared (**A**) and unpaired *t* test (**B**) using GraphPad Prism. * $P < 0.05$, ** $P < 0.01$, **** $P < 0.0001$; ns indicates not significant. Source data can be found in S1 Data.
(TIF)

**S11 Fig. A model of the comparison between mutation locations in *T. gondii* and *P. falciparum*.** *T. gondii* profilin (magenta) and actin (cyan) aligned with *P. falciparum* profilin (pink) and actin (blue) showing the similarity of the heterodimeric complex and the positions of the MMV291 *P. falciparum* mutations. The X-ray structure of *T. gondii* profilin (PDB: 3NEC) [85] and a homology model of *T. gondii* actin (created by SWISS-MODEL [106] using the X-ray structure of *P. falciparum* actin (PDB: 6I4K) [42] were used to create the model. The X-ray structure of *O. cuniculus* actin and human profilin (PDB: 2PBD) [56] was utilised as a template to spatially overlay the *P. falciparum* actin and profilin in the heterodimer model.
(TIF)

**S12 Fig. MMV291 series show limited activity against *T. gondii* invasion.** Nanoluciferase expressing parasites were liberated from their host cell and incubated with the MMV291 analogues before being added back to fibroblasts and allowed to invade for 1 hour before compounds were washed out. After a 24-hour incubation, cells were then lysed and the relative light units was quantified to correlate with *T. gondii* invasion rate. This showed MMV291 analogues *S*-W936 (**A**), *R*-W936 (**B**), *S*-MMV291 (**C**), and *R*-MMV291 (**D**) had some inhibitory activity against invasion at high concentrations. DMSO was included to the same amount as the highest concentration of analogue to account for DMSO-related effects (30% reduction in invasion). Error bars indicate the standard deviation from 2 biological repeats. Source data can be found in S1 Data.
(TIF)

**S13 Fig. MMV291 does not affect actin filaments in HeLa cells.** HeLa cells labelled with SiR-Actin imaged by lattice light-sheet microscopy upon stimulation with DMSO control (**A**), 5 μM Latrunculin B (**B**), 200 nM Cytochalasin D (CytD) (**C**), 2.5 μM MMV291 (**D**), 5 μM MMV291 (**E**), 10 μM MMV291 (**F**), and 20 μM MMV291 (**G**). Images represent a

100 × 100 μm subregion of a larger 250 × 250 μm field of view. Images are presented as maximum intensity projections with the contrast scaled between 100–400 counts. The images show the same region of cells imaged across multiple time points. Time is presented as HH:MM, and the scale bar represents 20 μm.
(TIF)

**S1 Movie. MMV291 does not affect actin filaments in HeLa cells.** HeLa cells labelled with SiR-Actin imaged by lattice light-sheet microscopy upon stimulation with DMSO Control, 5 μM Latrunculin B, 200 nM CytD, 2.5 μM MMV291, 10 μM MMV291, and 20 μM MMV291 over a time course of 3 hours. Movies represent maximum intensity projections with the contrast scaled between 100–400 counts. Time is presented as HH:MM, and the scale bar represents 20 μm.
(MP4)

**S1 Table. Primer sequences used to construct donor plasmids.**
(DOCX)

**S1 Data. Source data for graphs in this paper.**
(XLSX)

**S1 Raw Images. Uncropped images of gels in this paper.**
(PDF)

## Acknowledgments

We acknowledge the Australian Red Cross Blood Bank for the provision of human blood. We thank D. Marapana for the p1.2 CRISPR plasmid. We thank S. Tan for the PbCSP antibody and J. Boddey for the PfCSP antibody.

## Author Contributions

**Conceptualization:** Madeline G. Dans, William Nguyen, Christopher J. Tonkin, Geoffrey I. McFadden, Danny W. Wilson, Tania F. de Koning-Ward, Brad E. Sleebs, Inari Kursula, Paul R. Gilson.

**Data curation:** Madeline G. Dans, Henni Piirainen, Sachin Khurana, Somya Mehra, Zahra Razook, Niall D. Geoghegan, Aurelie T. Dawson, Sujaan Das, Molly Parkyn Schneider, Thorey K. Jonsdottir, Mikha Gabriela, Maria R. Gancheva, Vanessa Mollard, Christopher Dean Goodman.

**Formal analysis:** Madeline G. Dans, Henni Piirainen, Sachin Khurana, Somya Mehra, Zahra Razook, Niall D. Geoghegan, Molly Parkyn Schneider, Thorey K. Jonsdottir, Mikha Gabriela, Vanessa Mollard.

**Funding acquisition:** Brendan S. Crabb, Brad E. Sleebs, Paul R. Gilson.

**Investigation:** Madeline G. Dans, William Nguyen, Sachin Khurana, Sujaan Das, Brad E. Sleebs.

**Methodology:** Madeline G. Dans, Niall D. Geoghegan, Aurelie T. Dawson, Sujaan Das, Maria R. Gancheva, Christopher J. Tonkin, Vanessa Mollard, Christopher Dean Goodman, Geoffrey I. McFadden, Danny W. Wilson, Kelly L. Rogers, Alyssa E. Barry, Tania F. de Koning-Ward, Inari Kursula, Paul R. Gilson.

**Project administration:** Madeline G. Dans, Kelly L. Rogers, Brad E. Sleebs, Paul R. Gilson.

**Resources:** William Nguyen.

**Supervision:** Christopher J. Tonkin, Geoffrey I. McFadden, Danny W. Wilson, Kelly L. Rogers, Alyssa E. Barry, Brendan S. Crabb, Tania F. de Koning-Ward, Brad E. Sleebs, Inari Kursula, Paul R. Gilson.

**Writing – original draft:** Madeline G. Dans, Henni Piirainen, Paul R. Gilson.

**Writing – review & editing:** Madeline G. Dans, Henni Piirainen, William Nguyen, Sachin Khurana, Somya Mehra, Zahra Razook, Sujaan Das, Molly Parkyn Schneider, Thorey K. Jonsdottir, Mikha Gabriela, Maria R. Gancheva, Christopher J. Tonkin, Vanessa Mollard, Christopher Dean Goodman, Geoffrey I. McFadden, Danny W. Wilson, Alyssa E. Barry, Brendan S. Crabb, Tania F. de Koning-Ward, Brad E. Sleebs, Inari Kursula, Paul R. Gilson.

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
