## [Editor Report · Decision Letter 0]

7 Nov 2022

Dear Dr. Dans, 

Thank you for submitting your manuscript entitled "The sulfonylpiperazine MMV020291 prevents red blood cell invasion by the malaria parasite Plasmodium falciparum through interference with actin-1/profilin dynamics" for consideration as a Research Article by PLOS Biology.

Your manuscript has now been evaluated by the PLOS Biology editorial staff, as well as by an academic editor with relevant expertise, and I am writing to let you know that we would like to send your submission out for external peer review.

Once your full submission is complete, your paper will undergo a series of checks in preparation for peer review. After your manuscript has passed the checks it will be sent out for review. To provide the metadata for your submission, please Login to Editorial Manager (https://www.editorialmanager.com/pbiology) within two working days, i.e. by Nov 09 2022 11:59PM.

Kind regards,

Paula

---

Senior Editor

PLOS Biology

---

## [Decision Letter · Decision Letter 1]

12 Dec 2022

Dear Dr Dans,

Thank you for your patience while your manuscript "The sulfonylpiperazine MMV020291 prevents red blood cell invasion by the malaria parasite Plasmodium falciparum through interference with actin-1/profilin dynamics" was peer-reviewed at PLOS Biology. It has now been evaluated by the PLOS Biology editors, an Academic Editor with relevant expertise, and by several independent reviewers. 

In light of the reviews, which you will find at the end of this email, we would like to invite you to revise the work to thoroughly address the reviewers' reports.

As you will see below, the reviewers find your work novel and interesting. We consider that adding work to figure the exact mechanism out would improve the work, but we acknowledge that it can be a lot of work. If not possible to add all the work regarding the mechanism, we think that toning down the statements and conclusions and acknowledging the limitations of the study will be important for publication. Please address all the reviewers' concerns. 

Given the extent of revision needed, we cannot make a decision about publication until we have seen the revised manuscript and your response to the reviewers' comments. Your revised manuscript is likely to be sent for further evaluation by all or a subset of the reviewers.

**IMPORTANT - SUBMITTING YOUR REVISION**

*Re-submission Checklist*

*Published Peer Review*

*PLOS Data Policy*

*Blot and Gel Data Policy*

Sincerely,

Paula

---

Senior Editor

PLOS Biology

REVIEWS:

Reviewer #1: Cellular microbiology of protozoan parasites.

Reviewer #2: Cytoskeleton of Apicomplexa.

Reviewer #3: Parasite cell biology.

Reviewer #1: This is a convincing study that implicates actin-profilin dynamics in the mechanism of action of a potent compound series that blocks invasion of RBCs by Plasmodium falciparum. The literature review is comprehensive and the data are largely supportive of the model proposed, with some limitations. Although it is clear that the compounds affect actin-profilin dynamics and disrupt F actin formation, it is less clear what the exact target of the compounds is. This could be rectified by either performing the additional experiments described below or softening the conclusions about the nature of the "target". In either case, the work is interesting and highly deserving of publication.

1) There is somewhat of a disconnect between the selection of resistant mutants, the identification of mutations in actin and profilin, and the conclusion that these proteins are the "targets" of MMV291. My interpretation of "target" is that the compound binds directly to the molecule in question to alter its function, but I am not sure that is shown here. The inhibitor clearly alters the profilin-actin dynamics in vitro, and the mutants confer resistance in vivo, yet it is unclear if both molecules are "targets" in a classical sense. Data from the sedimentation data would support a direct interaction with the compounds and actin (since they increase turnover) but there is not a similar assay to show direct interaction with PFN. Hence, it is possible the mutations in PFN simply alter interactions with actin, which is the direct target. The authors favor a model where the mutations near the binding interface alter compound binding, but this is not shown directly. As well, the mutant PFN(K124) points away from the interface, so it must act allosterically. The authors should revise the text to be clear about what conclusions are really supported by the data and where they are speculating about mechanism. This issue affects the abstract, end of the introduction lines 222-223, and discussion.

2) The use of chromobodies to study polymerized actin is useful for monitoring the sensitivity to actin disrupting agents. However, this approach is somewhat artificial as it is almost certain that introduction of the chrombody alters actin dynamics and result in a more stable F actin pool in the parasite. This limitation should be acknowledged.

3) The results of the in vitro sedimentation assays are open to alternative interpretations. Addition of the MMV analogs to sedimented actin decreases the amount of F actin from 80% to 68% in one case, which is statistically significant. Addition of the compounds to a mixed fraction containing actin and PFN also further decreases the amount of F actin. The authors interpret this result to show that the compounds stabilize the PFN-G-Actin interaction, thus resulting in less F actin. This result might be better supported by examining the ATP-ADP exchange rate of actin when bound to PFN in the absence vs presence of compounds. Is there a direct effect on the stability of PFN-G-actin? Alternatively, it seems equally plausible that the compounds increase the turnover of F actin, releasing G actin which is then bound by PFN. Sorting out these two models would be possible using more quantitative assays for actin turnover (pyrene actin polymerization followed by compound addition and monitoring of stable actin) or for PFN nucleotide exchange, as suggested above. In the absence of more definitive data, the authors should modify their conclusions to allow for both possibilities. In particular, the Conclusion rather strongly states that there is no effect on actin polymerization directly, but the data on Figure 5 do not support this conclusion.

Minor comments:

Line 95 reference not formatted 

Line 114 MoF define

Reviewer #2: With 241 million cases and around 627 000 deaths, 77% of them concerning children aged under 5 years, malaria remains a devastating parasitic disease. Treatment failure due to emergence of artemisinin partial resistance increases the urgency for the development of novel parasiticidal drugs and more specifically drugs directed against new targets and with new mode of actions (MoA).

In this article Dans and colleagues characterize a series of new compounds targeting for the first time the actin-1/profilin complex and inhibiting the invasion of the red blood cell by the merozoite. This target was identified through the whole genome sequencing of chemically induced MMV291-resistant parasites and confirmed via reverse genetics. The MoA of the inhibitors was then dissected using biochemical approaches.

Overall, the manuscript is well-written, the experiments are clearly explained and well conducted with all the appropriate controls. The study is of high quality and of interest for a general audience since it identifies a new MoA. MMV291-based compounds stabilizes the interaction between Pfactin-1 and Pfprofilin, leading to a decrease of actin polymerization. In Apicomplexan parasites, actin filaments are made at the time of movement and are critical for invasion. Surprisingly the compound series is not acting on Toxoplasma gondii or other stages of Plasmodium falciparum.

I just have some issues concerning the delayed death phenotype experiments which I may not have understood correctly (Fig.6B, iii). How do the authors discriminate between an invasion phenotype and delayed death phenotype? In both cases, the biomass measured by nanoluciferase protein content would drop. Is it just because the drug was removed before merozoite invasion? Did the authors check the apicoplast morphology by immunofluorescence in the 2nd and 3rd cycle after a 40 h treatment stopped prior to merozoite invasion for 1 or 2 doses of MMV291?

It is somehow surprising not to have a stronger drop in the biomass if we compared to WT parasites of Fig. S1. The two experiments seem to be done with the same range of MMV291 concentrations but with a different assay (LDH vs nanoluciferase). Is it only due to different incubation time in presence of the compound?

Reviewer #3: In this manuscript by Das and colleagues, the authors identified actin and profilin as putative targets of MMV291. 

To identify these targets the authors generated MMV291 resistant parasites and identified 3 clonal lines with a mutation in actin (1 clone) or Profilin (2 clones). Replacing the endogenous genes with mutant variants, conserved resistance to MMV291.

The authors then undertake a series of experiments to investigate how MMV291 and analogs alter F-actin/profilin interaction and function. To summarize briefly the results - In vitro experiments with recombinantly purified Act1 and PFN demonstrate that PFN decreases the amount of F-actin in the same. The amount of F-actin is further reduced upon MMV291 treatment. In vivo, MMV291 alters actin organization in merozoites and apicoplast inheritance

These results are significant as they demonstrate that disruption of actin-pfn interaction is a potentially viable target for development of anti-malarial drugs. While this is an exciting prospect this claim is dependent on a MMV291 activity being specific for Plasmodium actin and pfn while not exhibiting inhibitor activity against human actin and profilin. This was not established by the authors. My second major concern is that authors repeatedly state that MMV291 increases actin sequestration by proflin but this is not directly demonstrated. 

Major concerns: 

1) The authors claim on a number of occasions (Abstract line 36; results line 346; discussion line 482) that the changes in actin organization and function after MMV291 treatment is due to increased actin-PFN binding. The authors should show this directly using their previously established native gel assay (Moreau et al 2017). Use of more sophisticated actin polymerization assays such as pyrene-polymerization assays previously used by the authors or TIRF microscopy assay would yield greater insight into how MMV291 is altering actin nucleation and elongation. 

2) MMV291 activity was not shown to be specific for plasmodium actin and profilin. In vitro polymerization assays with purified mammalian actin and profilin should be used to determine the specificity of the compounds. 

3) Chromobody images are low resolution and out of focus. Authors should provide images of higher resolution

4) Authors need to include detail microscopy methods for chromobody imaging - Minimum information needed includes microscope make and model, camera make and model, lens information (magnification and NA), and filter sets. 

5) Toxoplasma invasion assay is novel and not standard in the field. Authors should provide key controls that extracellular parasites are effectively washed off from the monolayers and are not significantly contributing to the assay. Secondly, the parasites invaded host cells in the presence of compounds. Then compounds were washed out and parasites grown for 24 hours post invasion. Authors should state if the inhibitory effect of these compounds are reversible and include appropriate references. If the effects are irreversible, authors should provide controls to demonstrate whether the compounds affect intracellular replication. 

Minor Concerns: 

Line 77: The definition of treadmilling is incorrect. 

Line78: ATP hydrolysis occurs once G-actin has been incorporated into the filament. As written G-actin-ATP to G-actin-ADP is misleading. This sentence should be re-written. 

Line 95: Moreau et al 2017 reference is not formatted

Line 114: Write out acronym MoA at first use

---

## [Editor Report · Decision Letter 2]

15 Feb 2023

Dear Dr. Dans,

Thank you for your patience while we considered your revised manuscript "The sulfonylpiperazine MMV020291 prevents red blood cell invasion by the malaria parasite Plasmodium falciparum through interference with actin-1/profilin dynamics" for publication as a Research Article at PLOS Biology. This revised version of your manuscript has been evaluated by the PLOS Biology editors and the Academic Editor.

Based on our Academic Editor's assessment of your revision, we are likely to accept this manuscript for publication, provided you address the following data and other policy-related requests.

1. DATA POLICY:

A) Supplementary files (e.g., excel). Please ensure that all data files are uploaded as 'Supporting Information' and are invariably referred to (in the manuscript, figure legends, and the Description field when uploading your files) using the following format verbatim: S1 Data, S2 Data, etc. Multiple panels of a single or even several figures can be included as multiple sheets in one excel file that is saved using exactly the following convention: S1_Data.xlsx (using an underscore).

B) Deposition in a publicly available repository. Please also provide the accession code or a reviewer link so that we may view your data before publication.

Regardless of the method selected, please ensure that you provide the individual numerical values that underlie the summary data displayed in the following figure panels as they are essential for readers to assess your analysis and to reproduce it: Figures 1B, 2B, 3ABCD, 4B, 5AB, 6AC, and supplementary figures S1, S2, S5AB, S7, S10AB, S12ABCD.

**Please also ensure that figure legends in your manuscript include information on where the underlying data can be found, and ensure your supplemental data file/s has a legend.**

2. Please provide a blurb which (if accepted) will be included in our weekly and monthly Electronic Table of Contents, sent out to readers of PLOS Biology, and may be used to promote your article in social media. The blurb should be about 30-40 words long and is subject to editorial changes. It should, without exaggeration, entice people to read your manuscript. It should not be redundant with the title and should not contain acronyms or abbreviations.

3. We propose a change in the title: "A sulfonylpiperazine compound prevents Plasmodium falciparum invasion of red blood cells through interference with actin-1/profilin dynamics".

We expect to receive your revised manuscript within two weeks.

*Published Peer Review History*

*Press*

Sincerely,

Paula

---

Senior Editor,

pjaureguionieva@plos.org,

PLOS Biology

---

## [Editor Report · Decision Letter 3]

6 Mar 2023

Dear Dr Dans,

Thank you for the submission of your revised Research Article "Sulfonylpiperazine compounds prevent Plasmodium falciparum invasion of red blood cells through interference with actin-1/profilin dynamics" for publication in PLOS Biology. On behalf of my colleagues and the Academic Editor, Kami Kim, I am pleased to say that we can in principle accept your manuscript for publication, provided you address any remaining formatting and reporting issues. These will be detailed in an email you should receive within 2-3 business days from our colleagues in the journal operations team; no action is required from you until then. Please note that we will not be able to formally accept your manuscript and schedule it for publication until you have completed any requested changes.

PRESS

Sincerely, 

Paula

---

Senior Editor

PLOS Biology
